# CMoS: Rethinking Time Series Prediction Through the Lens of Chunk-wise Spatial Correlations

Haotian Si [1 2]  Changhua Pei [1 3]  Jianhui Li [4 1]  Dan Pei [5]  Gaogang Xie [1]

## Abstract

Recent advances in lightweight time series forecasting models suggest the inherent simplicity of time series forecasting tasks. In this paper, we present CMoS, a super-lightweight time series forecasting model. Instead of learning the embedding of the shapes, CMoS directly models the spatial correlations between different time series chunks. Additionally, we introduce a Correlation Mixing technique that enables the model to capture diverse spatial correlations with minimal parameters, and an optional Periodicity Injection technique to ensure faster convergence. Despite utilizing as low as 1% of the lightweight model DLinear's parameters count, experimental results demonstrate that CMoS outperforms existing state-of-the-art models across multiple datasets. Furthermore, the learned weights of CMoS exhibit great interpretability, providing practitioners with valuable insights into temporal structures within specific application scenarios.

## 1. Introduction

Time series forecasting plays a vital role in many fields including finance, energy, and weather. By accurately predicting future time steps, it helps organizations make informed decisions and optimize resource allocation. As data-driven insights become increasingly essential, several forecasting methods based on deep learning are proposed with the interaction of structures like RNN (Lin et al., 2023), CNN (Hewage et al., 2020; LIU et al., 2022), and Transformers (Nie et al., 2023; Liu et al., 2024a). However, recent studies have shown that lightweight models can even outperform their complex counterparts (Zhang et al., 2022; Zeng et al., 2023; Das et al., 2023; Xu et al., 2024). This prompts us to reconsider a fundamental question: *whether temporal structures inherently possess a simple but efficient representation that we have previously overlooked.*

Several previous models (Nie et al., 2023; Wang et al., 2024) emphasized learning the representation of shapes, which is also called embeddings, to further build abstract dependency via attention or mixing mechanism. However, we argue that directly modeling the relative positional relationships between different time series chunks can be more robust and interpretable than modeling specific patterns. As shown in Fig. 1, for the subsequences of the sliding window, *while their shapes vary over time, as long as their relative positions are consistent, their dependencies, referred to as **chunk-to-chunk spatial correlations**, often maintain stable and reveal the temporal regularity of the system.* This phenomenon carries both analytical and practical implications: from an analytical perspective, it reveals that there's an inherent property with translation-equivariance in time series data, which can be a robust feature used for forecasting; from a practical perspective, we can employ relatively simple parametric methods for modeling such property, thereby substantially reducing model complexity.

[1]Computer Network Information Center, CAS, Beijing, China [2]University of Chinese Academy of Sciences, Beijing, China [3]Hangzhou Institute for Advanced Study, University of Chinese Academy of Sciences, Hangzhou, China [4]School of Frontier Sciences, Nanjing University, Nanjing, China [5]Tsinghua University, Beijing, China. Correspondence to: Jianhui Li <lijh@nju.edu.cn>.

*Proceedings of the 42$^{nd}$ International Conference on Machine Learning*, Vancouver, Canada. PMLR 267, 2025. Copyright 2025 by the author(s).

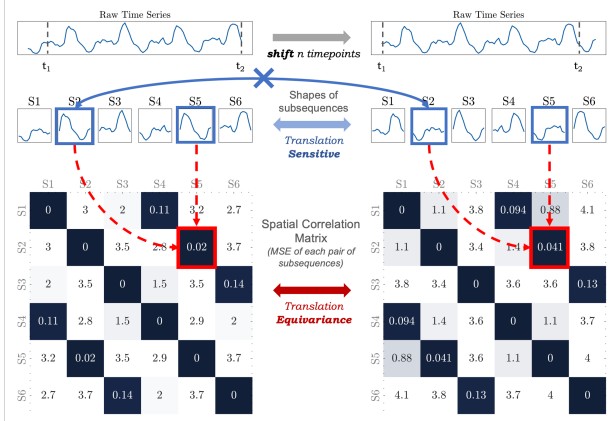

Figure 1. As time advances, the specific patterns in the time window change greatly, while the spatial correlations of the time series chunks remain similar.

Moreover, we find that such spatial correlation has good decomposability, which offers a new perspective for building differentiated spatial correlations in multivariate time series with much fewer parameters. Although the spatial correlations across different time series may be different, there could be inherent relationships between these correlations. For example, in power demand forecasting, industrial load exhibits long-term dependencies as it reflects stable consumption patterns shaped by long-term economic structural change, while residential demand shows predominantly short-term dependencies due to its sensitivity to immediate factors like weather conditions or public events. As a result, the total power consumption of this region demonstrates a hybrid dependency structure. Clearly, the spatial correlations in these three metrics are different from each other. However, as shown in Fig. 2, by decomposing the correlation dependencies into long-term parts and short-term parts, we can represent the spatial correlations of all three series simultaneously with only two sub-correlation matrices. Also, since each matrix is shared by multiple time series, the learned spatial correlation is less sensitive to noise and outliers within individual series.

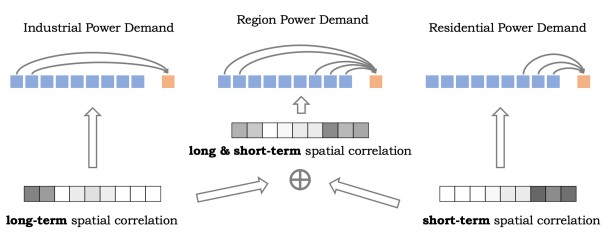

*Figure 2.* The spatial correlations of multiple time series can be represented by the combination of fewer sub-correlations.

In this paper, we pioneer leveraging the stability and decomposability of spatial correlations to build a super-lightweight forecasting model for multivariate time series. Specifically, we propose **CMoS**, a **C**hunk-wise **M**ixture **o**f **S**patial correlations architecture to predict multivariate time series. The architecture involves several techniques to investigate the robust and efficient representation of time series dependencies: (I) *Chunk-wise Spatial Correlation Modeling*. Instead of learning the representation of specific patterns, CMoS focuses on directly building the concise spatial correlation matrices for time series, and we prove both theoretically and experimentally that chunk-wise correlations exhibit stronger noise resistance than point-wise ones. (II) *Correlation Mixing Strategy*. CMoS introduces a mixture of correlation mechanisms to adaptively generate correlation structures across different time series. By learning a small set of basis correlation matrices and the corresponding mixing weights, CMoS can represent varying spatial relationships across channels while maintaining great parameter efficiency. (III) *Periodicity Injection by Weight*

*Editing*. Due to the high interpretability of the correlation matrices to be learned, we can inject periodicity into CMoS by directly editing the initial weights. This makes CMoS more easier to model the periodic spatial correlations, thereby speeding up convergence and enhancing the performance for time series with great periodicity. Empowered by the above techniques, CMoS can achieve state-of-the-art prediction performance with as low as **1% of the parameter count** compared to the lightweight model *DLinear*. Our code is provided in the anonymous repository `https://github.com/CSTCloudOps/CMoS`.

In summary, our contributions are as follows:

- We propose **CMoS**, a super-lightweight forecasting method with up to $100\times$ parameter efficiency than DLinear.

- Benefiting from chunk-wise spatial correlation modeling, correlation mixing strategy across channels, and Periodicity Injection techniques, CMoS achieves top-tier performance on long-term multivariate time series forecasting tasks.

- The learned spatial correlation matrices are highly interpretable, which can help us to better understand the potential temporal regularity of real-world systems.

## 2. Related Works

### 2.1. Development of lightweight time series forecasting models

Several previous works (Zhou et al., 2021; Wu et al., 2021; Zhou et al., 2022; Nie et al., 2023; Liu et al., 2024a) have adapted the transformer structure to the time series forecasting domain due to its capability of capturing long-term dependencies. The parameter count of these models often reaches the scale of tens of millions. However, DLinear (Zeng et al., 2023) demonstrated that merely using simple linear layers with fewer parameters can lead to competitive or even superior prediction performance, revealing that complex models with large parameter sizes are not always necessary to achieve high-quality forecasts. This has significantly accelerated the development of lightweight models, including FITS (Xu et al., 2024), SparseTSF (Lin et al., 2024b), and CycleNet (Lin et al., 2024a), to achieve state-of-the-art prediction performance with fewer parameters.

The success of the lightweight models inspires a crucial insight: there likely exists an elegantly simple yet highly effective formulation for the structure of time series within the existing prediction framework (predicting future time steps based on lookback windows). Thus we try to shift our focus from modeling sophisticated pattern representations to directly learning the inherent and interpretable spatial

correlations among all channels, and finally design the super-lightweight model CMoS.

## 2.2. Channel Strategy for Lightweight Models

Multivariate time series can be viewed as a multi-channel signal. Whether it is necessary to model the dependencies between different channels has been widely discussed. Some works claimed that modeling these cross-channel dependencies can enhance the prediction results (Chen et al., 2023; Han et al., 2024; Liu et al., 2024a), while some thought that modeling all channels separately with only one backbone (Nie et al., 2023; Xu et al., 2024), which is known as *Channel Independent Strategy*, can provide more robustness and further lead to better results. Since the latter does not require the additional overhead associated with modeling cross-channel dependencies, this greatly decreases the overall parameter count. As a result, lightweight models (Xu et al., 2024; Lin et al., 2024b) whose parameter counts are comparable to or smaller than DLinear tend to adopt this Channel Independent strategy.

In this work, we reexamine the drawbacks of the Channel Independent strategy in the design of lightweight models from the perspective of model capacity. For lightweight models, the scarcity of nonlinear relationships inherently restricts their ability to express diverse temporal structures. Therefore, employing the channel-independent strategy will confine the model to representing only one single temporal structure. In CMoS, we address this issue by introducing Correlation Mixing, which significantly enhances the model's capacity to express a variety of temporal structures with an affordable parameter cost. Additionally, we provide a detailed comparison of the modeling overhead associated with different channel strategies and the predictive performance of CMoS variants in the Appendix B, further highlighting the superiority of our Correlation Mixing strategy.

## 3. CMoS

In this section, we introduce each component employed in CMoS and demonstrate their advantages.

### 3.1. Chunk-wise Spatial Correlation Modeling

Time series forecasting is the task of predicting future values of a sequence $\{x_1, x_2, \ldots, x_t\}$ based on its past observations. Given a sequence of $L$ observations, the goal is to predict the next value(s) $x_{t+1}, x_{t+2}, \ldots, x_{t+H}$, where $H$ is the forecasting horizon. Formally, the task can be defined as:

$$\hat{x}_{t+1}, \ldots, \hat{x}_{t+H} = f(\{x_{t-L+1}, x_{t-L+2}, \ldots, x_t\}, \boldsymbol{\theta}) \quad (1)$$

where $f$ is the forecasting function, $\boldsymbol{\theta}$ represents the model parameters, and $\{x_{t-L+1}, x_{t-L+2}, \ldots, x_t\}$ is the window

of past $L$ observations.

Motivated by the translation-equivariance of chunk-to-chunk spatial correlations mentioned in Sec. 1, we redefine the forecasting task as the following simple formulation. A time series $\{x_{t-L+1}, x_{t-L+2}, \ldots, x_t\} \in \mathbb{R}^L$ with length $L$ can be divided into the chunk-wise series $\{\mathbf{x}_{t-L+1:t-L+S}, \mathbf{x}_{t-L+S+1:t-L+2S}, \ldots, \mathbf{x}_{t-S+1:t}\} \in \mathbb{R}^{\frac{L}{S} \times S}$ with length $\frac{L}{S}$ when the chunk size is $S$, and we denote it as $\{\mathbf{x}_{t-\frac{L}{S}+1}, \mathbf{x}_{t-\frac{L}{S}+2}, \ldots, \mathbf{x}_t\}$ in brief. For each chunk $\mathbf{x}_{t+i}$ to be predicted, from the perspective of spatial correlations, it can be simply viewed as a linear combination of previous sequences:

$$\mathbf{x}_{t+i} = \sum_{j=0}^{\frac{L}{S}} \theta_{ij} \mathbf{x}_{t-j} + \mathbf{b}_i \quad (2)$$

where the learnable $\theta_{ij}$ denotes the spatial correlation coefficient, indicating the extent to which the $\frac{L}{S} - j$-th chunk in the historical window influences the prediction of the $i$-th chunk. The learnable $\mathbf{b}_i$ can introduce some basic trends like gradual increase.

In fact, some existing lightweight models, such as DLinear and TSMixer, can be viewed as special cases with a chunk size of 1, where they model *point-to-point spatial correlations*. However, we find that this type of correlation is more susceptible to random noise. We demonstrate that chunk-to-chunk spatial correlation can be more robust to random noise than point-to-point spatial correlation.

**Definition 3.1.** Consider a linear regression model $f(\mathbf{x}; \boldsymbol{\theta}) = \boldsymbol{\theta}^\top \mathbf{x}$. Assume the model is subjected to input $\mathbf{x}' = \mathbf{x} + \delta$, where $\delta \sim \mathcal{N}(\mu, \sigma^2)$ is a Gaussian noise vector. We define the *noise sensitivity of the model* as the variance of the output change: $Var(f(\mathbf{x}'; \boldsymbol{\theta}) - (f(\mathbf{x}; \boldsymbol{\theta})) = Var(\boldsymbol{\theta}^\top \delta) = \sigma^2 \|\boldsymbol{\theta}\|_2^2$.

**Theorem 3.2.** *Within each chunk, perform a weighted average of the point-to-point weights $\{\theta_1, \theta_2, \ldots, \theta_n\}$ to obtain new weights $\theta^* = \frac{\sum_{i=1}^n \alpha_i \theta_i}{\sum_{i=1}^n \alpha_i} (\alpha_i \geq 0)$. Consequently, we have $\sigma^2 \sum_{i=1}^n \theta_i^2 \geq \sigma^2 \theta^{*2}$, i.e., the chunk-wise linear model composed of new weights exhibit lower noise sensitivity under Definition 3.1.*

The proof of **Theorem 3.2** is provided in Appendix F.

**Chunking v.s. Patching.** The Patching technique, proposed by Nie et al. (2023), is widely used in many existing time series forecasting methods (Goswami et al., 2024; Woo et al., 2024; Liu et al., 2024b). Technically, patching splits only the historical series into segments, and patch-based models focus on generating aggregated representation of the correlations between these historical segments (similar to the high-level semantic information in LLMs) and then decode the representation to future time points. However, the black-box nature of such representations make it hard to figure out

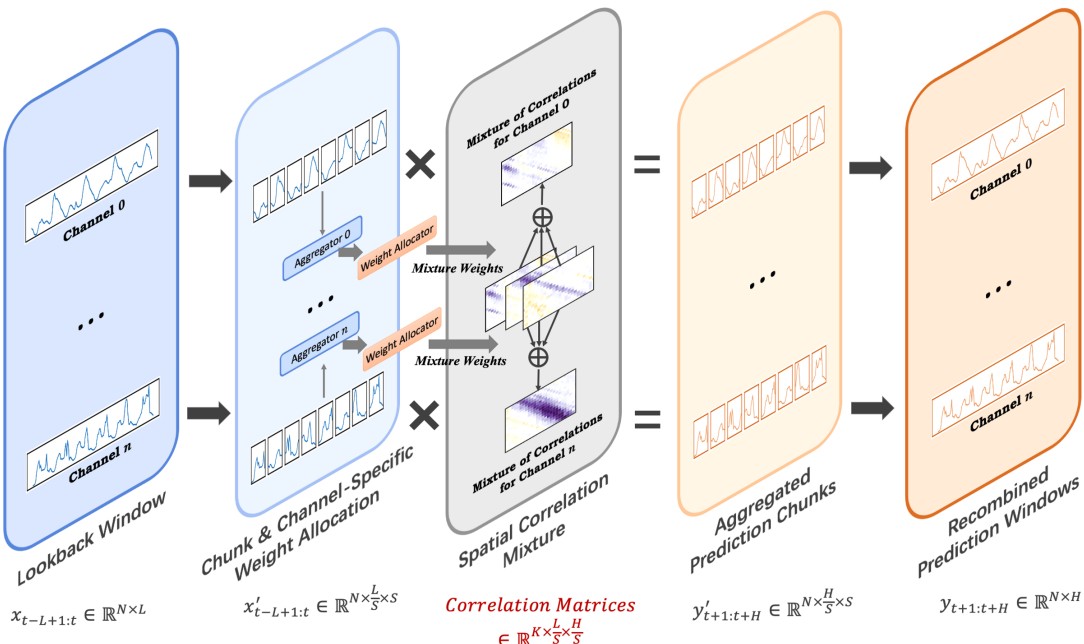

*Figure 3.* **CMoS** Architecture.

how specific segment influence the final prediction, limiting the interpretability of these methods.

In contrast, the proposed chunking technique splits both historical and future series, and instead of learning the high-level representations, chunk-based CMoS focus on directly modeling of the spatial correlation between historical and future segments. which is quite interpretable. Our further experiments also demonstrate that chunks enhance both robustness and efficiency.

### 3.2. Correlation Mixing

Most recent works either adopt the Channel Independent strategy which is potentially incompatible with multiple temporal structures when parameter count is limited, or adopt the channel-mixing strategy while introducing unaffordable complexity. To address the above shortcuts, we design the correlation mixing strategy to better model the spatial correlations meanwhile maintaining great parameter efficiency. As a simplified version of Mixture-of-Expert (Jacobs et al., 1991) (each expert is just a spatial correlation matrix), this strategy combines the various basic spatial correlations with weighted sums obtained by the aggregated information of the lookback windows of each channel, adaptively generating channel-specific spatial correlations. The strategy includes three components: shared spatial correlation matrices, channel-specific information aggregators, and a shared weight allocator.

**Shared spatial correlation matrices.** Inspired by the

decomposability of the spatial correlations, we expand the sole chunk-wise correlations to the combination of $K$ basic spatial correlations according to the weight set $\Gamma^n = \{\gamma_0^n, \dots, \gamma_{K-1}^n\}$ given chunk-wise lookback window of the $n$-th channel $\{\mathbf{x}_{t-\frac{L}{S}+1}^n, \mathbf{x}_{t-\frac{L}{S}+2}^n, \dots, \mathbf{x}_t^n\}$:

$$\mathbf{x}_{t+i}^n = \frac{1}{\sum_{k=0}^{K} e^{\gamma_k^n}} \sum_{k=0}^{K} e^{\gamma_k^n} \sum_{j=0}^{\frac{L}{S}} (\theta_{ij}^k \mathbf{x}_{t-j}^n + \mathbf{b}_i^k) \quad (3)$$

and the $K$ correlation matrices $\boldsymbol{\theta}^0, \dots, \boldsymbol{\theta}^{K-1}$ are shared by all channels.

**Two-stage weight allocation.** The key to building robust spatial correlation matrices as well as obtaining the best correlation for each channel is establishing a stable mapping between raw time series data and weights of each matrix. In our formulation, channels with similar temporal structures should have similar weight combinations. However, due to the potential differences in noise levels across time series data from different channels, the weight allocation process may still be affected by the noise level, even if their temporal structures are similar. Thus the first stage of weight allocation is to reduce the effect raised by different noise levels. Specifically, **for each channel**, we design a *specific information aggregator* via a convolution kernel with size $c$ and stride $\frac{c}{2}$. This operation applies the specific degree of smoothing to the raw data of a certain channel to obtain its stable and sparse representation $\mathbf{z} \in \mathbb{R}^{\frac{2L-c}{c}}$:

$$\mathbf{z}_t^n = Conv1D^n(\mathbf{x}_{t-L+1:t}^n) \quad (4)$$

In the second stage, since the effect of the noise level has been mitigated, which means that channels with similar temporal structures have a similar representation, we employ a linear layer (the parameter matrix $\in \mathbb{R}^{\frac{2L-c}{c} \times K}$) **shared by all channels** as a *weight allocator* to map the sparse representation $\mathbf{z}_t^n$ to the weight set $\Gamma^n = \{\gamma_0^n, \ldots, \gamma_{K-1}^n\}$:

$$\Gamma^n = Linear(\mathbf{z}_t^n) \tag{5}$$

which is used for combining correlation matrices in Eq. (3).

### 3.3. Periodicity Injection by Weight Editing

Some recent works (Lin et al., 2024b;a) have shown that explicitly leveraging the periodic features of time series can effectively improve the prediction accuracy of the model. In CMoS, given a periodic time series with the main period $p$, the time series chunks to be predicted should be more closely related to the corresponding chunks of previous periods, *i.e.*, $Correlation(\mathbf{x}_t, \mathbf{x}_{t-i}) \gg Correlation(\mathbf{x}_t, \mathbf{x}_{t-j})$ for $\forall i, j$ such that $i \mod \frac{p}{S} = 0$ and $j \mod \frac{p}{S} \neq 0$. As a result, the spatial correlation matrix will exhibit a significant peak every $\frac{p}{S}$ chunks.

According to this assumption, we can directly edit the parameters in the first spatial correlation matrix by *initializing it with the pre-defined periodic peaks*. Specifically, the $\theta_{ij}^{edit}$ in the matrix $\boldsymbol{\theta}^{edit}$ that satisfy $\frac{L}{S} - i + j \mod \frac{p}{S} = 0$ is assigned to $\frac{p}{L}$, while others are assigned to 0, as shown in Fig 4. For time series data with significant periodicity, since the initialized parameters already incorporate most periodic correlations, the model's learning burden is reduced. By providing a strong inductive bias towards periodicity, this operation not only accelerates convergence but also enhances model robustness. Meanwhile, the remaining correlation matrices are freed to focus on capturing non-periodic temporal structures, which can lead to an improvement in final prediction performance. In CMoS, we apply this strategy to those datasets with human activities involvement (demonstrating periodicity that aligns with human calendar cycles, *e.g.*, daily or weekly routines), and calculate the period of these datasets by AutoCorrelation Function (Madsen, 2007). More details about Periodicity Injection are provided in Appendix E.

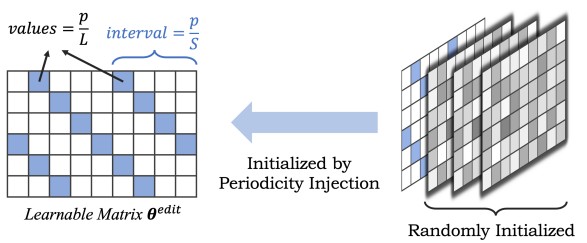

$values = \frac{p}{L}$     $interval = \frac{p}{S}$

*Learnable Matrix $\boldsymbol{\theta}^{edit}$*

Initialized by
Periodicity Injection

Randomly Initialized

*Figure 4.* Illustration of Periodicity Injection.

### 3.4. Instance Normalization

We employ the Reversible Instance Normalization (Kim et al., 2022) technique, which is commonly used in many previous works, to address the potential distribution shift issue and improve prediction performance. Specifically, we normalize the original data and add statistical features back to the model's output to obtain the original distribution:

$$\mathbf{x}_{t-L+1:t}^n = \frac{\mathbf{x}_{t-L+1:t}^n - \mu}{\sqrt{\sigma}} \tag{6}$$

$$\hat{\mathbf{x}}_{t+1:t+H}^n = \hat{\mathbf{x}}_{t+1:t+H}^n \times \sqrt{\sigma} + \mu \tag{7}$$

where $\mu$ and $\sigma$ denote the mean and the variance of the input window.

### 3.5. Loss Function

We apply the Mean Squared Error (MSE) of the predicted values and the ground truth values as the loss function $\mathcal{L}$:

$$\mathcal{L} = \frac{1}{N} \sum_{n=0}^{N} \|\hat{\mathbf{x}}_{t+1:t+H}^n - \mathbf{x}_{t+1:t+H}^n\|_2^2 \tag{8}$$

### 3.6. Parameter Efficiency Analysis

Suppose the length of the lookback window is $L$ and the forecast horizon is $H$ for $N$ channels. Even with an overly naive strategy of sharing only one temporal structure across all channels, the total parameter count of $DLinear$ is higher than $2 \times L \times H$. In CMoS, given chunk size $S$, convolution kernel size $c$, and number of basic correlation matrices $K$ ($K$ is a constant and $K \ll N$). The total number of parameters in CMoS is:

$$\underbrace{K \times \frac{L}{S} \times \frac{H}{S}}_{Correlation\ Part} + \underbrace{N \times c}_{Aggregators\ Part} + \underbrace{\frac{2L-c}{c} \times K}_{Weight\ Allocator\ Part}$$

For the dataset ETTh1 with $N = 7$, $S$ is set to 24, $K$ is set to 4, and $c$ is set to 8, the total number of the CMoS's parameters $\approx \frac{L \times H}{96} + L$, which is less than 1% of DLinear's parameter count, while CMoS can additionally deal with multiple temporal structures.

## 4. Experiments

In this section, we present both the performance and parameter efficiency advantages of CMoS against existing state-of-the-art baselines, and demonstrate the effectiveness of each design in CMoS. Additional experimental results on hyperparameter sensitivity are listed in Appendix C.2.

### 4.1. Setup

**Datasets.** We conduct experiments on 7 widely used datasets for long-term time series forecasting, including

*Table 1.* The prediction performance of CMoS and baselines on long-term multivariate time series forecasting task. The results are averaged from all prediction horizons of $H \in \{96, 192, 336, 720\}$. The best result is highlighted in **bold** and the second best is highlighted with underline. The results of CMoS are averaged over 5 runs on each horizon with standard deviation **STD**. For detailed results on each prediction horizon, please refer to Appendix C.1.

| | Datasets | Electricity | | Traffic | | Weather | | ETTh1 | | ETTh2 | | ETTm1 | | ETTm2 | |
|---|---|---|---|---|---|---|---|---|---|---|---|---|---|---|---|
| | Metrics | MSE | MAE | MSE | MAE | MSE | MAE | MSE | MAE | MSE | MAE | MSE | MAE | MSE | MAE |
| Complex Models | Informer | 0.389 | 0.406 | 1.898 | 0.866 | 0.292 | 0.348 | 0.739 | 0.589 | 0.422 | 0.440 | 0.584 | 0.503 | 0.362 | 0.387 |
| | FedFormer | 0.219 | 0.329 | 0.62 | 0.382 | 0.301 | 0.345 | 0.433 | 0.454 | 0.406 | 0.438 | 0.567 | 0.519 | 0.334 | 0.380 |
| | TimesNet | 0.190 | 0.284 | 0.617 | 0.327 | 0.255 | 0.282 | 0.468 | 0.459 | 0.390 | 0.417 | 0.408 | 0.415 | 0.292 | 0.331 |
| | PatchTST | 0.171 | 0.270 | 0.397 | 0.275 | 0.224 | 0.261 | 0.429 | 0.436 | 0.351 | 0.395 | **0.349** | 0.381 | 0.256 | 0.314 |
| | TimeMixer | 0.185 | 0.284 | 0.410 | 0.279 | 0.226 | 0.264 | 0.427 | 0.441 | 0.350 | 0.397 | 0.356 | 0.380 | 0.258 | 0.318 |
| | iTransformer | 0.163 | 0.258 | 0.397 | 0.281 | 0.232 | 0.270 | 0.439 | 0.448 | 0.370 | 0.403 | 0.361 | 0.390 | 0.269 | 0.327 |
| Lightweight Models | DLinear | 0.167 | 0.264 | 0.428 | 0.287 | 0.242 | 0.295 | 0.430 | 0.443 | 0.470 | 0.468 | 0.356 | 0.378 | 0.259 | 0.324 |
| | FITS | 0.168 | 0.265 | 0.429 | 0.302 | 0.244 | 0.281 | 0.408 | 0.427 | 0.335 | 0.386 | 0.357 | **0.377** | 0.254 | 0.313 |
| | SparseTSF | 0.165 | 0.258 | 0.412 | 0.278 | 0.240 | 0.280 | 0.406 | 0.419 | 0.344 | **0.364** | 0.361 | 0.382 | **0.251** | 0.312 |
| | CycleNet | **0.158** | **0.250** | 0.421 | 0.289 | 0.242 | 0.278 | 0.415 | 0.426 | 0.355 | 0.398 | 0.355 | 0.379 | 0.252 | **0.309** |
| | CMoS | **0.158** | **0.250** | **0.396** | **0.271** | **0.220** | **0.260** | **0.403** | **0.416** | **0.331** | 0.383 | 0.354 | 0.378 | 0.259 | 0.316 |
| | **STD** | ±0.001 | ±0.001 | ±0.001 | ±0.001 | ±0.002 | ±0.002 | ±0.004 | ±0.003 | ±0.003 | ±0.002 | ±0.003 | ±0.003 | ±0.004 | ±0.004 |

Electricity , Traffic, Weather, ETTh1, ETTh2, ETTm1, and ETTm2. More details of datasets are listed in Appendix D.1.

**Baselines.** We compare CMoS with several well-known and state-of-the-art methods, including Informer (Zhou et al., 2021), FedFormer (Zhou et al., 2022), TimesNet (Wu et al., 2023), PatchTST (Nie et al., 2023), TimeMixer (Wang et al., 2024), iTransformer (Liu et al., 2024a), and lightweight models like DLinear (Zeng et al., 2023), FITS (Xu et al., 2024), SparseTSF (Lin et al., 2024b), and CycleNet (Lin et al., 2024a). More implementation details about baselines are provided in Appendix D.4.

**Experimental Settings.** Following previous lightweight methods like FITS and SparseTSF, we conduct a grid search on the lookback window of 96, 336, 720. For each setting, the results of CMoS are averaged over 5 runs with random seeds. More setting details about CMoS are listed in Appendix D.3.

### 4.2. Main Results

Table 1 presents a comprehensive comparison of prediction results between CMoS and other baseline models across multiple datasets. Overall, CMoS demonstrates superior predictive performance, achieving state-of-the-art results on the majority of benchmarks, ranking first in 9 out of 14 evaluation metrics and second in 3 metrics. It is noteworthy that CMoS consistently achieves optimal results on datasets containing more than 20 channels (Eletricity, Traffic, and Weather). Interestingly, on these datasets, for methods employing the channel independence strategy, complex models (PatchTST, TimeMixer, iTransformer) generally outperform the simpler models by a significant margin. The reason lies in that under this strategy, the representation capacity

of complex models is sufficient enough, while lightweight models are limited to modeling only one temporal structure (*e.g.*, spatial correlation). As a result, they struggle to effectively handle multiple time series with varying temporal structures at the same time. Owing to the mixture of correlations mechanism, CMoS is able to deal with various temporal structures inherent in numerous time series samples, and even surpasses the prediction performance of complex models.

Furthermore, CMoS exhibits remarkable stability in performance across varying random seed initializations, which can be attributed to the minimal parameter count and multiple robustness-enhancing strategies, particularly the Chunk-wise Modeling and Periodicity Injection mechanisms. The combined effect of these design choices notably contributes to the model's reproducibility and practical deployment capabilities.

### 4.3. Ablation Study

We validate the effectiveness of each technique in CMoS through multiple ablation studies as follows.

**Effectiveness of Chunk-wise Correlation Modeling.** To investigate the effectiveness of chunk-wise modeling against point-wise modeling, we compare the prediction performance of CMoS and its variant, CMoS w/o Chunk, implemented by setting the chunk size to 1. Even though the parameter size of models using chunk-wise modeling is typically only one-tenth to one-hundredth of that of models using point-wise modeling, according to Table 2, the former outperforms the latter in terms of prediction performance on the majority of datasets, and the improvement is particularly notable on the ETT-series datasets which exhibit

higher noise levels. This suggests that chunk-wise correlation modeling is an approach with both greater parameter efficiency and robustness.

**Effectiveness of Correlation Mixing.** To evaluate the impact of the Correlation Mixing mechanism, we implement a variant of CMoS by restricting the number of spatial correlation matrices to one (CMoS w/o Correlation Mixing). In this simplified configuration, all channels are constrained to share a single spatial correlation, which is equivalent to adopting the channel independence strategy. Our experimental results demonstrate that this restriction significantly impacts model performance. Consistent with our comprehensive comparison against other lightweight models presented in Sec. 4.2, CMoS with Correlation Mixing exhibits substantial performance advantages over its variants, and this superiority is especially pronounced in datasets containing more channels (achieving remarkable improvements of up to 17.5% on the Weather dataset). These results provide compelling evidence for the effectiveness of our correlation mixing strategy and its crucial role in representing various temporal structures while maintaining model efficiency.

*Table 2.* The MSE and MAE results of CMoS and its variants on 7 datasets with horizon = 96. CorMix is the abbreviation of Correlation Mixing. The best results are highlighted in **bold**, and the worst results are marked with *.

| Variants | CMoS | | w/o CorMix | | w/o Chunk | |
|---|---|---|---|---|---|---|
| Metrics | MSE | MAE | MSE | MAE | MSE | MAE |
| Electricity | **0.129** | **0.223** | 0.137* | 0.231* | 0.132 | 0.226 |
| Traffic | **0.367** | **0.256** | 0.385* | 0.266* | 0.368 | 0.259 |
| Weather | **0.144** | **0.193** | 0.171* | 0.226* | 0.147 | 0.203 |
| ETTh1 | **0.361** | **0.383** | 0.362 | 0.385 | 0.368* | 0.393* |
| ETTh2 | **0.274** | 0.341 | 0.276 | **0.338** | 0.280* | 0.344* |
| ETTm1 | 0.292 | 0.345 | 0.308 | 0.349 | **0.288** | **0.340** |
| ETTm2 | **0.167** | 0.257 | 0.168 | **0.257** | 0.170* | 0.262* |

**Effectiveness of Periodicity Injection.** The implementation of Periodicity Injection enables the model to more efficiently learn spatial correlations associated with primary periodic patterns, resulting in substantial improvements in the speed of loss reduction, as shown in Fig.5.

Furthermore, this mechanism provides strong prior knowledge about periodic structures, allowing the remaining parameters to concentrate on capturing secondary periodicities and non-periodic components of the time series. This, as shown in Table 3, can potentially enhance overall performance in most cases. However, from the experimental results, we observe an exception in the Weather dataset. Unlike other datasets that are dominated by human activities and are highly correlated with human-made cyclical patterns, the Weather dataset is derived from natural phe-

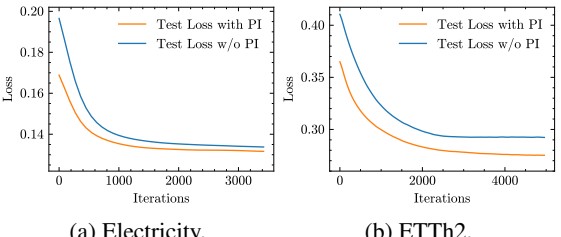

(a) Electricity.        (b) ETTh2.

*Figure 5.* Test loss during training process on different datasets with horizon= 96. PI is the abbreviation of Periodicity Injection. The model with Periodicity Injection exhibits faster loss reduction.

nomena and is influenced by more random and complex factors. As a result, the periodicity of most metrics in this dataset is either not imprecise or not significant. Therefore, for such datasets, Periodicity Injection may have a negative impact. In practice, we recommend using this strategy only when significant and stable periodicity exists in the majority of the time series.

*Table 3.* The MSE results of CMoS with Periodictiy Injection and w/o Periodicity Injection.

| | Electricity | Traffic | Weather | ETTh1 | ETTh2 | ETTm1 | ETTm2 |
|---|---|---|---|---|---|---|---|
| with PI | 0.129 | 0.367 | 0.148 | 0.361 | 0.274 | 0.292 | 0.167 |
| w/o PI | 0.130 | 0.369 | 0.144 | 0.366 | 0.292 | 0.293 | 0.170 |

### 4.4. Effectiveness and Efficiency

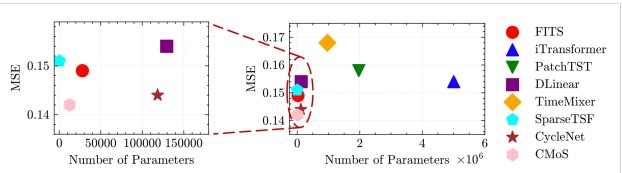

*Figure 6.* Comparison of the prediction performance and parameter count between CMoS and other baselines on Electricity dataset with horizon= 192.

A key advantage of CMoS lies in its ability to achieve superior predictive performance with remarkably few parameters. As demonstrated in our experimental results on the Electricity dataset in Fig. 6, CMoS achieves state-of-the-art prediction accuracy while utilizing only a fraction of the parameters required by most existing lightweight models - even one to two orders of magnitude fewer. This demonstrates that our modeling approach, combining spatial correlation modeling and correlation mixing, though concise, effectively captures temporal structures in forecasting tasks. This reveals the potentially inherent low-rank nature of time series prediction. Furthermore, such a compact parametrization enables CMoS to be readily deployed

on edge devices, enabling high-quality time series forecasting even in resource-constrained environments. Meanwhile, to further verify the efficiency of CMoS in practice, we provide the inference FLOPs, GPU memory footprint, and inference time of CMoS and other baselines on Electricity dataset on our platform in Appendix C.3.

## 5. Interpretability

In the CMoS model, each weight $\theta_{i,j}$ in the spatial correlation matrix $\theta$ directly reflects the strength of the relationship between the corresponding historical chunk and the predicted chunk. A larger value of $\theta_{i,j}$ in the matrix indicates that the model places greater emphasis on the i-th historical chunk when predicting the j-th future chunk. To gain deeper insights into the model's ability to learn and capture these spatial correlations, as shown in Fig. 7, we visualize four spatial correlation matrices, which are also referred to as correlation mappings, that are learned from the weather dataset without periodicity injection.

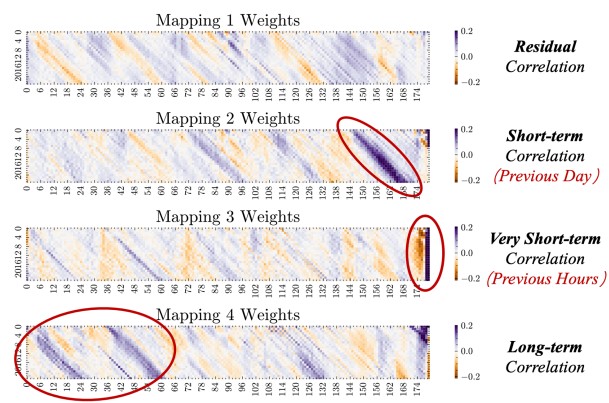

*Figure 7.* Visualization of the learned spatial correlation matrices (mappings) on the Weather dataset. The chunk size is set to 4.

The visualization analysis reveals that each spatial correlation mapping captures and emphasizes distinct patterns of temporal dependencies, which can help understand the inherent patterns of the whole system. We explain the features of each mapping one by one:

**Mapping 2**. Under the setting with a chunk size of 4, the weights in Mapping 2 exhibit a very distinct diagonal stripe pattern at the tail end, where each weight $\theta_{i,j}$ on this stripe roughly satisfies the condition $180 - i + j = 36$. Considering that the weather dataset is sampled every ten minutes ($6 \times 24/4 = 36$ chunks per day), this stripe indicates that the predicted chunk is highly dependent on the historical data from the same chunk one day prior. In other words, *Mapping 2 primarily models the short-term dependency between the predicted value and its observation from the previous day*.

**Mapping 3**. In contrast, the larger weights in Mapping 3 are almost entirely concentrated at the very end of the matrix. This indicates that *the mapping models very short-term dependencies, relying heavily on observations from the past hour or even the last few minutes during prediction*. This is a more effective forecasting strategy for time series where long-term trends are unpredictable but short-term trends remain relatively stable.

**Mapping 4**. Mapping 4, on the other hand, exhibits several noticeable diagonal stripes near the beginning of the matrix, suggesting that the prediction of a particular chunk depends more on data from other chunks observed a long time ago. Therefore, *Mapping 4 models long-term dependencies, which are likely to be more prominent in time series with strong and stable periodic features*.

**Mapping 1**. The weights in Mapping 1 are distributed relatively evenly, without showing very strong or specific dependencies. We hypothesize that *the model uses Mapping 1 to capture finer-grained dependencies that other mappings overlook, namely residual dependencies*.

To further explore how the model combines these foundational correlations to perform prediction on each channel, we visualize the original time series along with the average mapping allocation situation corresponding to each time series as shown in Fig. 8 (for all cases please refer to Appendix H). As previously discussed, Channel 1 exhibits slow and unpredictable trend variations, prompting the model to exclusively utilize Mapping 3 to capture very short-term dependencies. Channel 3 also exhibits some slow trends, but a peak occurs every other day, and the shape of the peak is more similar to that of nearby peaks. Consequently, the model primarily combines Mapping 2 and Mapping 3 to jointly model the very short-term and cross-day dependencies. Channel 4 demonstrates more distinct periodic characteristics, leading the model to additionally leverage the long-term dependencies captured in Mapping 4 to enhance prediction performance and robustness.

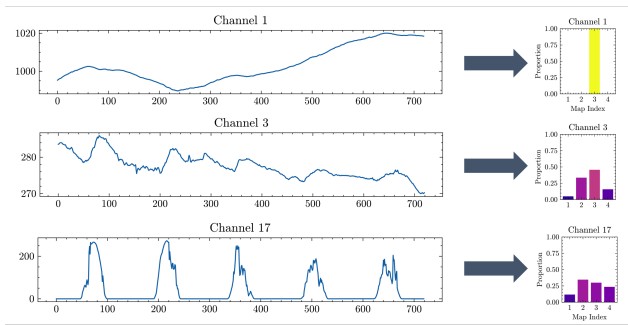

*Figure 8.* Raw time series and their corresponding averaged mapping allocation proportion.

# 6. Conclusion

In this paper, we propose a super-lightweight time series forecasting model **CMoS**, which directly builds the spatial correlation within different time series chunks. To enhance the model's capacity to model varying temporal structures, we propose the Correlation Mixing strategy to adaptively combine the foundational correlations for one specific channel. Also, we introduce the Periodicity Injection trick to accelerate the convergence. The experimental results show that CMoS achieves top-tier prediction performance with an extremely limited number of parameters. Additionally, our further analysis demonstrates that the model has excellent interpretability, which helps us better understand the underlying patterns of the system.

# Acknowledgements

This work was partially funded by the National Natural Science Foundation of China (Grant No. W2412136), the National Key Research and Development Program of China (No.2022YFB2901800), the National Natural Science Foundation of China (62202445), and the National Natural Science Foundation of China-Research Grants Council (RGC) Joint Research Scheme (62321166652).

# Impact Statement

This paper presents work whose goal is to advance the field of Machine Learning. There are many potential societal consequences of our work, none which we feel must be specifically highlighted here.

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

## A. CMOS Circuit & Our CMoS Model

The design of our model, which coincidentally shares the CMoS nomenclature, partly draws some inspiration from CMOS (Complementary Metal-Oxide-Semiconductor) circuits. There exists an interesting parallel between the two: just as CMOS circuits dynamically switch their outputs between VDD (high voltage level) and GND (ground level) in response to varying input signals, our model adaptively combines different spatial correlations in response to various channels within the lookback window, as shown in Fig. 9.

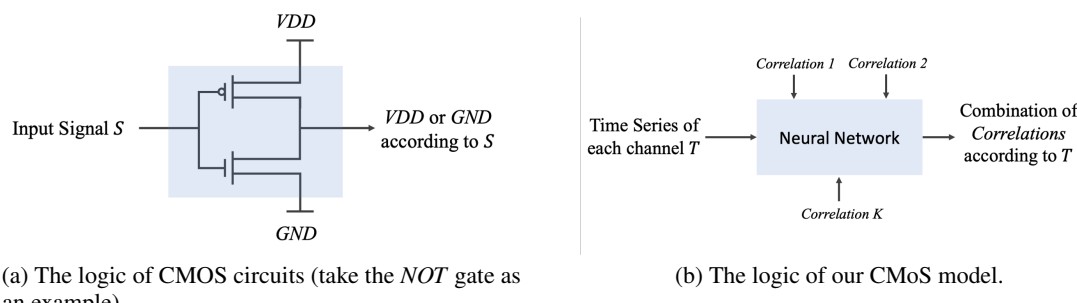

(a) The logic of CMOS circuits (take the *NOT* gate as an example).

(b) The logic of our CMoS model.

*Figure 9.* The logical similarity between CMOS circuits and CMoS model.

## B. More Information about Channel Strategy

### B.1. Classification of Channel Strategies

We summarize the channel strategies that existing methods applied as follows (and shown in Fig. 10):

- **Private Line**. This strategy is adopted by some methods that decompose multivariate time series forecasting task to multiple univariate time series forecasting tasks (Box & Pierce, 1970; Oreshkin et al., 2020). In this mode, each channel is modeled completely separately, with each channel having its own dedicated network. For a dataset with $N$ channels, this results in a model complexity of $O(N)$, and may suffer from overfitting since the training data of each network is not enough.

- **One Bus**. This strategy is also called *Channel-Independence* in several works (Nie et al., 2023; Liu et al., 2024b; Goswami et al., 2024; Lin et al., 2024b). Each channel can only receive past information from its own, and all channels share only one temporal structure. Although this strategy is claimed to be able to avoid overfitting since the network is trained with more data, a unified structure may struggle to handle the diverse temporal structures when faced with different time series from the same system.

- **Data-Mixing**. This strategy, adopted by several recent works (Zhang & Yan, 2023; Chen et al., 2023; Liu et al., 2024a; Han et al., 2024), assumes that data from the other channels can benefit the predictions for one specific channel. However, this will result in the model complexity of $O(N^2)$ for building cross-channel dependencies, making it a cumbersome solution that is not suitable for many cases, especially for edge devices.

- **Correlation-Mixing**. This strategy is proposed by CMoS. We posit that channel dependencies manifest through similar spatial correlations across different channels. By mixing $K$ ($K \ll N$) fundamental correlations, we can derive the unique temporal structure for each channel while maintaining great parameter efficiency.

Table 4 shows the model complexity when using different channel strategies with the same backbone (assuming the complexity of the backbone is constant $C$) on a dataset containing $N$ channels.

### B.2. Experimental Results on Different Channel Strategy

We further compare CMoS with its variants under these channel strategies by removing or rewriting the correlation mixing module. For the data-mixing strategy, since the model with complexity of $O(CN^2)$ exceeds the GPU memory limit when the number of channels is large (more than 100), for comprehensive analysis, we chose TSMixer, a model based on Multilayer

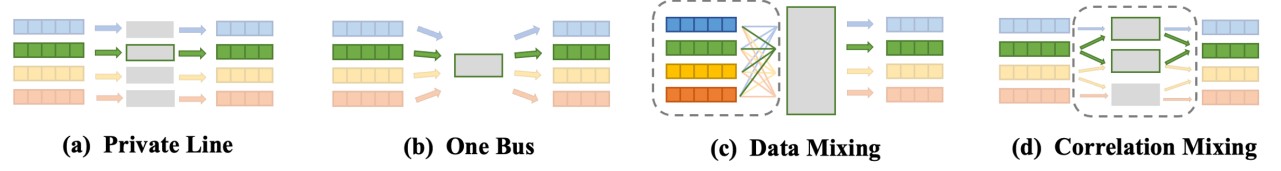

**(a) Private Line**   **(b) One Bus**   **(c) Data Mixing**   **(d) Correlation Mixing**

*Figure 10.* Illustration of different channel strategies.

*Table 4.* The model complexity under different channel strategies. The complexity of the backbone is constant $C$, and the number of channel is $N$.

| Strategy | Private Line | One Bus | Data Mixing | Correlation Mixing |
|---|---|---|---|---|
| **Complexity** | $O(CN)$ | $O(C)$ | $O(CN^2)$ | $O(CK), K \ll N$ |

Perceptron (MLP) architecture that incorporates simplified data mixing techniques (the model's parameter count is still several hundred times that of CMoS), as our benchmark model under data-mixing strategy. Table 5 includes the forecasting results under all channel strategies. One Bus strategy, as mentioned above, will limit the capacity of the model, while the Private Line strategy will limit the number of training samples for each separate network, both would degrade the performance. Although the Data Mixing strategy brings heavy complexity, its performance still does not surpass that of CMoS. We will discuss this intriguing phenomenon in the next section.

*Table 5.* Predicition performance under different channel strategies. All datasets include $> 20$ channels.

| Strategy | Private Line | | One Bus | | Data Mixing | | Correlation Mixing | |
|---|---|---|---|---|---|---|---|---|
| Metric | MSE | MAE | MSE | MAE | MSE | MAE | MSE | MAE |
| Electricity | 0.135 | 0.229 | 0.137 | 0.231 | 0.131 | 0.229 | **0.129** | **0.223** |
| Traffic | 0.399 | 0.281 | 0.385 | 0.266 | 0.376 | 0.264 | **0.367** | **0.256** |
| Weather | 0.148 | 0.206 | 0.171 | 0.226 | 0.147 | 0.203 | **0.144** | **0.193** |

### B.3. Cross-channel Dependency from the Perspective of Information Theory

Contrary to the assumption of Data Mixing strategy that building cross-channel needs additional modules compared with models using Private Line strategy, from the perspective of information theory, we believe that cross-channel dependencies should not lead to an increase in model complexity, but rather provide greater opportunities for model compression. Consider the Shannon Entropy for the $i$-th channel $H(X^i) = -\sum P(x^i) \log P(x^i)$, the Shannon Entropy for the $j$-th channel $H(X^j) = -\sum P(x^j) \log P(x^j)$, and their joint entropy $H(X^i, X^j) = -\sum\sum P(x^i, x^j) \log P(x^i, x^j)$. The existence of cross-channel dependency means that $X^i$ provides some information for $X^j$. Under the information theory framework, this means the mutual information $I(X^i, X^j)$ of the two channels is greater than 0. Since $I(X^i, X^j) = H(X^i) + H(X^j) - H(X^i, X^j)$, we have $H(X^i, X^j) < H(X^i) + H(X^j)$ if the cross-channel independency exists. The Private Line strategy can be seen as a modeling of $\sum H(X^i)$. Therefore, in the presence of cross-channel dependencies, **we assume that fewer parameters are required to model the mapping from the historical window to the prediction window**, as the joint information entropy of the data $H(X^1, \ldots, X^N)$ is smaller than $\sum H(X^i)$.

## C. More Results

### C.1. Full Results Compared with Baselines

The full results compared with baselines with horizons $\in \{96, 192, 336, 720\}$ are shown in Table 6. Compared to all state-of-the-art methods, CMoS maintains top-2 prediction accuracy in most settings.

*Table 6.* The full prediction performance of CMoS and baselines on long-term multivariate time series forecasting task with horizons ∈ {96, 192, 336, 720}. The best result is highlighted in **bold** and the second best is highlighted with underline. The results of CMoS are averaged over 5 runs on each horizon.

| | Datasets | Electricity | | Traffic | | Weather | | ETTh1 | | ETTh2 | | ETTm1 | | ETTm2 | |
|---|---|---|---|---|---|---|---|---|---|---|---|---|---|---|---|
| | Metrics | MSE | MAE | MSE | MAE | MSE | MAE | MSE | MAE | MSE | MAE | MSE | MAE | MSE | MAE |
| *Horizon = 96* | Informer | 0.215 | 0.321 | 0.682 | 0.391 | 0.206 | 0.253 | 0.715 | 0.571 | 0.362 | 0.394 | 0.419 | 0.422 | 0.216 | 0.302 |
| | FedFormer | 0.191 | 0.305 | 0.593 | 0.365 | 0.175 | 0.242 | 0.379 | 0.419 | 0.337 | 0.380 | 0.463 | 0.463 | 0.216 | 0.309 |
| | TimesNet | 0.169 | 0.271 | 0.595 | 0.312 | 0.168 | 0.214 | 0.389 | 0.412 | 0.334 | 0.370 | 0.340 | 0.378 | 0.189 | 0.265 |
| | PatchTST | 0.143 | 0.247 | 0.370 | 0.262 | 0.149 | 0.196 | 0.377 | 0.397 | **0.274** | **0.337** | **0.289** | **0.343** | 0.165 | 0.255 |
| | TimeMixer | 0.153 | 0.256 | 0.369 | **0.256** | 0.147 | 0.198 | 0.372 | 0.401 | 0.281 | 0.351 | 0.293 | 0.345 | 0.165 | 0.256 |
| | iTransformer | 0.134 | 0.230 | **0.363** | 0.265 | 0.157 | 0.207 | 0.386 | 0.405 | 0.297 | 0.348 | 0.300 | 0.353 | 0.175 | 0.266 |
| | DLinear | 0.140 | 0.237 | 0.395 | 0.275 | 0.170 | 0.230 | 0.379 | 0.403 | 0.300 | 0.364 | 0.300 | 0.345 | 0.164 | 0.255 |
| | FITS | 0.139 | 0.237 | 0.400 | 0.280 | 0.172 | 0.225 | 0.376 | 0.396 | 0.277 | 0.345 | 0.303 | 0.345 | 0.165 | 0.254 |
| | SparseTSF | 0.138 | 0.233 | 0.389 | 0.268 | 0.169 | 0.223 | 0.362 | 0.388 | 0.294 | 0.346 | 0.312 | 0.354 | 0.163 | 0.252 |
| | CycleNet | **0.129** | **0.223** | 0.397 | 0.278 | 0.167 | 0.221 | 0.374 | 0.396 | 0.279 | 0.341 | 0.299 | 0.348 | **0.160** | **0.247** |
| | CMoS | **0.129** | **0.223** | 0.367 | **0.256** | **0.144** | **0.193** | **0.361** | **0.383** | **0.274** | 0.341 | 0.292 | 0.345 | 0.167 | 0.257 |
| *Horizon = 192* | Informer | 0.263 | 0.362 | 2.802 | 1.275 | 0.261 | 0.300 | 0.726 | 0.574 | 0.460 | 0.448 | 0.547 | 0.480 | 0.320 | 0.365 |
| | FedFormer | 0.203 | 0.316 | 0.614 | 0.381 | 0.274 | 0.344 | 0.420 | 0.444 | 0.415 | 0.428 | 0.575 | 0.516 | 0.297 | 0.360 |
| | TimesNet | 0.180 | 0.280 | 0.613 | 0.322 | 0.219 | 0.262 | 0.440 | 0.443 | 0.404 | 0.413 | 0.392 | 0.404 | 0.254 | 0.310 |
| | PatchTST | 0.158 | 0.260 | 0.386 | 0.269 | 0.191 | 0.239 | 0.409 | 0.425 | 0.348 | 0.384 | **0.329** | 0.368 | 0.221 | 0.293 |
| | TimeMixer | 0.168 | 0.269 | 0.400 | 0.271 | 0.192 | 0.243 | 0.413 | 0.430 | 0.349 | 0.387 | 0.335 | 0.372 | 0.225 | 0.298 |
| | iTransformer | 0.154 | 0.250 | 0.384 | 0.273 | 0.200 | 0.248 | 0.424 | 0.440 | 0.372 | 0.403 | 0.341 | 0.380 | 0.242 | 0.312 |
| | DLinear | 0.154 | 0.250 | 0.407 | 0.280 | 0.216 | 0.275 | 0.427 | 0.435 | 0.387 | 0.423 | 0.336 | 0.366 | 0.224 | 0.304 |
| | FITS | 0.149 | 0.248 | 0.412 | 0.288 | 0.215 | 0.261 | **0.400** | 0.418 | **0.331** | 0.379 | 0.337 | **0.365** | 0.219 | 0.291 |
| | SparseTSF | 0.151 | 0.244 | 0.398 | 0.270 | 0.214 | 0.262 | 0.403 | 0.411 | 0.339 | **0.377** | 0.347 | 0.376 | 0.217 | 0.290 |
| | CycleNet | 0.144 | 0.237 | 0.411 | 0.283 | 0.212 | 0.258 | 0.406 | 0.415 | 0.342 | 0.385 | 0.334 | 0.367 | **0.214** | **0.286** |
| | CMoS | **0.142** | **0.236** | **0.379** | **0.261** | **0.186** | **0.237** | 0.405 | **0.409** | 0.333 | 0.383 | 0.334 | 0.366 | 0.228 | 0.299 |
| *Horizon = 336* | Informer | 0.334 | 0.416 | 0.881 | 0.496 | 0.309 | 0.332 | 0.741 | 0.588 | 0.454 | 0.464 | 0.654 | 0.531 | 0.400 | 0.414 |
| | FedFormer | 0.221 | 0.333 | 0.627 | 0.389 | 0.331 | 0.374 | 0.458 | 0.466 | 0.389 | 0.457 | 0.618 | 0.544 | 0.366 | 0.400 |
| | TimesNet | 0.204 | 0.293 | 0.626 | 0.332 | 0.278 | 0.302 | 0.523 | 0.487 | 0.389 | 0.435 | 0.423 | 0.426 | 0.313 | 0.345 |
| | PatchTST | 0.168 | 0.267 | **0.396** | 0.275 | 0.242 | **0.279** | 0.431 | 0.444 | 0.377 | 0.416 | **0.362** | 0.390 | 0.276 | 0.327 |
| | TimeMixer | 0.189 | 0.291 | 0.407 | 0.272 | 0.247 | 0.284 | 0.438 | 0.450 | 0.367 | 0.413 | 0.368 | 0.386 | 0.277 | 0.332 |
| | iTransformer | 0.169 | 0.265 | **0.396** | 0.277 | 0.252 | 0.287 | 0.449 | 0.460 | 0.388 | 0.417 | 0.374 | 0.396 | 0.282 | 0.337 |
| | DLinear | 0.169 | 0.268 | 0.417 | 0.286 | 0.258 | 0.307 | 0.440 | 0.440 | 0.490 | 0.487 | 0.367 | 0.386 | 0.277 | 0.337 |
| | FITS | 0.170 | 0.268 | 0.426 | 0.301 | 0.261 | 0.295 | 0.419 | 0.435 | 0.350 | 0.396 | 0.368 | **0.384** | 0.272 | 0.326 |
| | SparseTSF | 0.166 | 0.260 | 0.411 | 0.275 | 0.257 | 0.293 | 0.434 | 0.428 | 0.359 | **0.307** | 0.367 | 0.386 | 0.270 | 0.327 |
| | CycleNet | **0.161** | **0.254** | 0.424 | 0.289 | 0.260 | 0.293 | 0.431 | 0.430 | 0.371 | 0.413 | 0.368 | 0.386 | **0.269** | **0.322** |
| | CMoS | **0.161** | **0.254** | 0.397 | **0.270** | **0.240** | 0.281 | **0.412** | **0.420** | **0.342** | 0.384 | 0.366 | 0.386 | 0.273 | 0.325 |
| *Horizon = 720* | Informer | 0.502 | 0.525 | 3.225 | 1.302 | 0.390 | 0.388 | 0.772 | 0.623 | 0.410 | 0.454 | 0.715 | 0.578 | 0.512 | 0.468 |
| | FedFormer | 0.259 | 0.364 | 0.646 | 0.394 | 0.423 | 0.418 | 0.474 | 0.488 | 0.483 | 0.488 | 0.612 | 0.551 | 0.459 | 0.450 |
| | TimesNet | 0.206 | 0.293 | 0.635 | 0.340 | 0.353 | 0.351 | 0.521 | 0.495 | 0.434 | 0.448 | 0.475 | 0.453 | 0.413 | 0.402 |
| | PatchTST | 0.214 | 0.307 | **0.435** | **0.295** | 0.312 | **0.330** | 0.457 | 0.477 | 0.406 | 0.441 | **0.416** | 0.423 | 0.362 | 0.381 |
| | TimeMixer | 0.228 | 0.320 | 0.462 | 0.316 | 0.318 | 0.330 | 0.486 | 0.484 | 0.401 | 0.436 | 0.426 | 0.417 | 0.360 | 0.387 |
| | iTransformer | 0.194 | 0.288 | 0.445 | 0.308 | 0.320 | 0.336 | 0.495 | 0.487 | 0.424 | 0.444 | 0.429 | 0.430 | 0.375 | 0.394 |
| | DLinear | 0.203 | 0.300 | 0.454 | 0.308 | 0.324 | 0.367 | 0.473 | 0.494 | 0.704 | 0.597 | 0.419 | 0.416 | 0.371 | 0.401 |
| | FITS | 0.212 | 0.304 | 0.478 | 0.339 | 0.326 | 0.341 | 0.435 | 0.458 | 0.382 | 0.425 | 0.420 | **0.413** | 0.359 | 0.381 |
| | SparseTSF | 0.205 | 0.293 | 0.448 | 0.297 | 0.321 | 0.340 | **0.426** | 0.447 | 0.383 | 0.424 | 0.419 | **0.413** | **0.352** | **0.379** |
| | CycleNet | **0.198** | **0.287** | 0.450 | 0.305 | 0.328 | 0.339 | 0.450 | 0.464 | 0.426 | 0.451 | 0.417 | 0.414 | 0.363 | 0.382 |
| | CMoS | 0.200 | 0.288 | 0.442 | **0.295** | **0.311** | 0.332 | 0.433 | 0.451 | **0.374** | **0.423** | 0.425 | 0.417 | 0.367 | 0.385 |

## C.2. Parameter Sensitivity

We conduct experiments to investigate the impact of the hyperparameters *chunk size* and *the number of spatial correlation matrices* on the final prediction performance. Fig. 11a demonstrates the prediction results with different chunk sizes. Since the main period in the Electricity dataset is $24 \times 7 = 168$, when the chunk size is set to 16 which is not a common factor of the period, the MSE increases greatly. In practice, setting the chunk size as a common divisor of the potential periods would be a better strategy for CMoS.

Fig. 11b demonstrates the prediction results with different numbers of spatial correlation matrices. For the Electricity dataset which contains over 100 channels, when the number is small (1 and 2), the model fails to capture sufficient temporal dependencies, leading to suboptimal performance. However, more matrices bring more parameter count and more computational cost, and may lead to the network becoming overly sparse, which might also slightly affect the overall performance. Therefore, for datasets with a medium or large number of channels, it would be a better choice when the number of spatial correlation matrices is set to an integer between 4 and 8.

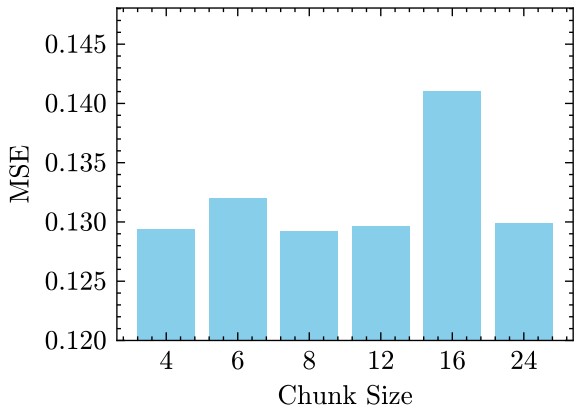
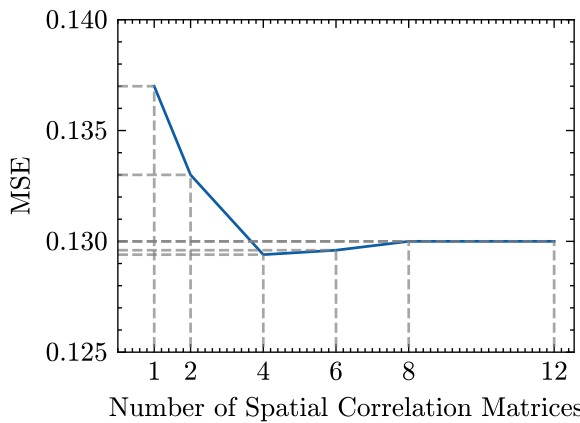

(a) MSE results with varied chunk size.

(b) MSE results with the varied number of spatial correlation matrices.

*Figure 11.* MSE results on Electricity dataset with multiple hyperparameter settings. The horizon is set to 96.

## C.3. Inference Efficiency

We additionally provide the inference FLOPs, GPU memory footprint, and inference time of CMoS and other baselines on Electricity dataset using a 3090 GPU as follows. The batch size of all methods is set to 64 for fair comparison.

*Table 7.* The inference FLOPs, GPU memory footprint, and inference time of CMoS and other baselines on Electricity dataset using one 3090 GPU.

| Method | DLinear | CycleNet | SparseTSF | FITS | iTransformer | PatchTST | TimeMixer | CMoS |
|---|---|---|---|---|---|---|---|---|
| FLOPS | 5.31G | 5.68G | 1.02G | 5.33G | 249.51G | 1196.08G | 10.58G | 2.96G |
| GPU Memory | 245MB | 267MB | 262MB | 691MB | 2271MB | 22014MB | 18642MB | 252MB |
| Inference Time | 1.81s | 1.83s | 1.49s | 4.71s | 1.92s | 2.90s | 2.85s | 1.58s |

# D. Detailed Experimental Settings

## D.1. Datasets

We train and test all methods on 7 well-known datasets that are widely used for long-term forecasting: Electricity, Traffic, Weather, ETTh1, ETTh2, ETTm1, and ETTm2. Since Christoph Bergmeir (2024) strongly questioned the reasonability of

another dataset called Exchange_rate on the NIPS'24 workshop, we excluded this dataset from our experiments. The number of time steps, number of channels, sample rate, the period obtained by AutoCorrelation Function, and the description of each dataset are listed in Table 8. Following most previous works (Zhou et al., 2022; Nie et al., 2023; Xu et al., 2024; Lin et al., 2024a), the ETT-series datasets are divided into training, validation, and test sets with a 6:2:2 ratio, while others are divided into training, validation, and test sets with a 7:1:2 ratio.

*Table 8.* The detailed information of datasets.

| Datasets | Time steps | Channels | Sample rate | Period obtained by ACF | Description |
|---|---|---|---|---|---|
| Electricity (2021) | 26304 | 321 | 1 hour | 168 | Electricity consumption data of 321 clients |
| Traffic (2021) | 17544 | 862 | 1 hour | 168 | Road occupancy rates measured by 862 sensors |
| Weather (2021) | 52696 | 21 | 10 min | 144 | 21 meteorological factors |
| ETTh1 (2021) | 17420 | 7 | 1 hour | 24 | 7 factors of electricity transformer |
| ETTh2 (2021) | 17420 | 7 | 1 hour | 24 | 7 factors of electricity transformer |
| ETTm1 (2021) | 69680 | 7 | 15 min | 96 | 7 factors of electricity transformer |
| ETTm2 (2021) | 69680 | 7 | 15 min | 96 | 7 factors of electricity transformer |

### D.2. Environment

All model instances in the paper are implemented with PyTorch (Paszke et al., 2019). We conducted all experiments on the server equipped with 12 Intel(R) Xeon(R) Gold 5317 CPUs @ 3.00GHz and 4 NVIDIA GA102 GeForce RTX 3090 GPUs. Each experiment is conducted with one GPU.

### D.3. Implementation Details of CMoS

Due to the simplicity of CMoS, we only need to set a few hyperparameters. In our experiment, one set of hyperparameters is searched for each dataset. The chunk size is searched from $\{2,4,8,24\}$, the number of spatial correlation matrices is searched from $\{2,4,8\}$, and the lookback window is searched from $\{96,336,720\}$. We use the AdamW (Loshchilov & Hutter, 2019) optimizer and the learning rate is searched from $\{2 \times 10^{-5}, 5 \times 10^{-5}, 8 \times 10^{-5}, 8 \times 10^{-4}\}$. We additionally adopt the StepLR scheduler with $stepsize = 20$ and $gamma = 0.75$. Each model is trained for 200 epochs with batch size= 64.

### D.4. Experimental Settings on other baselines

We reimplement CycleNet and SparseTSF according to their source code in `https://github.com/lss-1138/SparseTSF` and `https://github.com/ACAT-SCUT/CycleNet`. TFB (Qiu et al., 2024) searches the lookback window and other hyperparameters for all baselines, providing a comprehensive and reliable benchmark for these baselines. Therefore, for other baselines included in the latest version of TFB, we directly refer to the prediction performance provided by TFB.

## E. Details of Periodicity Injection

### E.1. Implementation Details of Periodicity Injection

For each dataset, firstly, we use the AutoCorrelation Function (ACF) to calculate the dominant period of this dataset (the period of all datasets calculated by ACF are listed in the 5th column in Table 8 in Appendix D.1). Next, during the grid search process for the hyperparameter chunk size, for each experiment setting, we input the the calculated and the hyperparameter chunk size into Algorithm 1 provided in Appendix E.2, to obtain a modified matrix initialized via Periodicity Injection. This matrix is then used to replace the first basic correlation matrix of the initialized model, thereby completing the Periodicity Injection operation.

### E.2. Pseudocode of Periodicity Injection

---

**Algorithm 1** Pseudocode of Periodicity Injection

---

**Input:** Zero-Initialized Spatial Correlation Matrix $\boldsymbol{\theta} \in \mathbb{R}^{\frac{H}{S} \times \frac{L}{S}}$, period $p$, chunk size $S$
**for** $i = 1$ **to** $\frac{H}{S}$ **do**
   **for** $j = \frac{L}{S} - \frac{p}{S}$ **to** $1$ **with step** $\frac{p}{S}$ **do**
      **if** $i + j < \frac{L}{S}$ **then**
         $\theta_{i,j+i} = \frac{p}{L}$
      **end if**
   **end for**
**end for**
**Return:** Modified Matrix $\boldsymbol{\theta}$

---

## F. Theoretical Proof of Theorem 3.2

*Proof.* To prove Theorem 3.2, we need to demonstrate that:

$$\left(\frac{\sum_{i=1}^{n} \alpha_i \theta_i}{\sum_{i=1}^{n} \alpha_i}\right)^2 \leq \sum_{i=1}^{n} \theta_i^2 (\alpha_i \geq 0) \tag{9}$$

Since $\alpha_i \geq 0$, we have $(\sum_{i=1}^{n} \alpha_i)^2 \geq \sum_{i=1}^{n} \alpha_i^2$. Thus, we have:

$$\left(\frac{\sum_{i=1}^{n} \alpha_i \theta_i}{\sum_{i=1}^{n} \alpha_i}\right)^2 \leq \frac{(\sum_{i=1}^{n} \alpha_i \theta_i)^2}{\sum_{i=1}^{n} \alpha_i^2} \tag{10}$$

According to the Cauchy-Schwarz inequality, we have:

$$(\sum_{i=1}^{n} \alpha_i \theta_i)^2 \leq \sum_{i=1}^{n} \alpha_i^2 \sum_{i=1}^{n} \theta_i^2 \tag{11}$$

Thus, we finally prove that:

$$\left(\frac{\sum_{i=1}^{n} \alpha_i \theta_i}{\sum_{i=1}^{n} \alpha_i}\right)^2 \leq \frac{(\sum_{i=1}^{n} \alpha_i \theta_i)^2}{\sum_{i=1}^{n} \alpha_i^2} \leq \sum_{i=1}^{n} \theta_i^2 (\alpha_i \geq 0) \tag{12}$$

The equality holds if and only if at most one $\alpha_i$ is non-zero ($i \in \{0, \ldots, n-1\}$). Therefore, the original theorem is proven.

## G. Further Discussion

Although CMoS has demonstrate its superior prediction performance and efficiency on existing datasets, we further discuss some theoretical aspects that are not yet fully developed.

**Spatial Correlation Modeling.** The premise of modeling spatial correlation is that the time series data exhibit certain relatively stable patterns, including but not limited to local smoothness, periodicity, and long-term trends. For those irregular time series, such as in the case of a time series generated by random walks, the effectiveness of this modeling approach has yet to be fully validated. Nevertheless, it is notable that periodicity or stationary trend constitutes a fundamental indicator of time series predictability and forecasting potential. Long-term forecasting can be very challenging if time series exhibit no periodicity or stationary trend.

**Extension of Theorem 3.2** Theorem 3.2 considers Gaussian noise, which is one of the most common and widely adopted assumptions. For those Non-Gaussian noise, take the burst noise as an example, it can be viewed as extreme deviations that occur in the tail of a noise distribution. So we can define burst noise $B(t)$ mathematically as follows based on extreme value theory. Let $X(t)$ be a noise process. According to the Pickands–Balkema–de Haan theorem, the conditional distribution of the exceedances events exceeding a sufficiently high threshold $u$ follow a Generalized Pareto Distribution (GPD):

$$GPD(y; \sigma, \xi) = P(Y(t) \leq y | X(t) > u) \approx 1 - \left(1 + \frac{\xi y}{\sigma}\right)^{-1/\xi}$$

where $y = x - u > 0$, $\sigma > 0$ is the scale parameter, and $\xi > 0$ is the shape parameter.

Next, since the burst noise can be regarded as discrete events, we define their occurence times $T_i$ follow a Poisson process with intensity $\lambda(u)$. So we can finally define burst noise $B(t)$ as

$$B(t) = y + u, \text{ if } t \in \{T_i\}, \text{ otherwise } 0$$

where $y$ is i.i.d. $GPD(\sigma, \xi)$.

However, when we attempt to replace $\delta$ with $B$ in Definition 3.1, we find it difficult to theoratically analyze $Var(\theta^T B)$ since B is sparse and thus $Var(\theta^T B)$ relies heavily on specific samples of GPD.

As an alternative, we choose to experimentally compare the performance of chunk/point-level modeling on the time series with random burst noise. Specifically, we construct several time series using sine functions, and following the above formulation, we additionally inject some burst noises into all time series. The prediction results are listed in Table 9, and we can conclude that chunk-level modeling is more robust than point-level modeling when facing burst noise, which can be seen as an empirical extension of Theorem 3.2 on other non-Gaussian noise like burst noise.

*Table 9.* Prediction performance on time series with burst noise.

|  | MSE | MAE |
| --- | --- | --- |
| Chunk-level | 0.0246 | 0.1166 |
| Point-level | 0.0249 | 0.1176 |

## H. Extended Visualization on the Weather Dataset

We provide more visualization on the Weather dataset to showcase more relationships between the raw time series and their corresponding mapping allocation proportions. Fig. 12 provides visualizations of all the raw time series channels within a long time window in the Weather dataset, and Fig. 13 provides visualizations of mapping allocation situation on all channels.

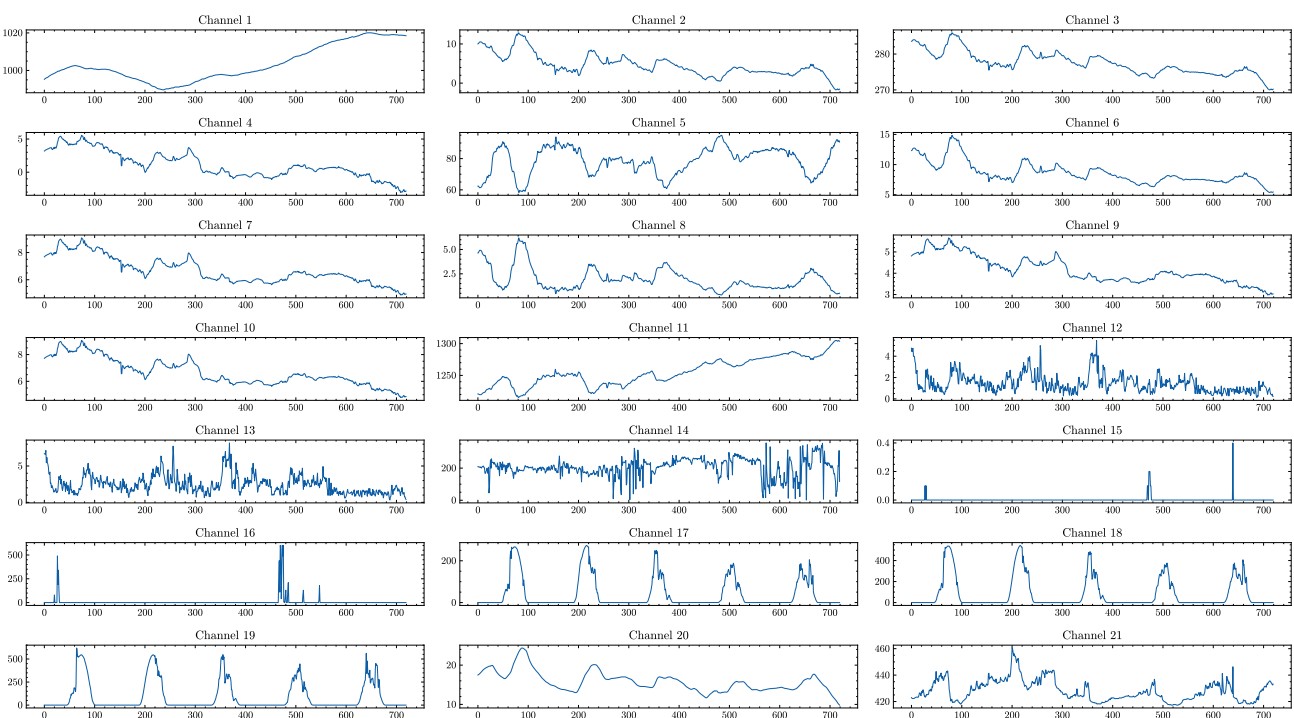

*Figure 12.* Visualizations of all channels within a long time window in the Weather dataset.

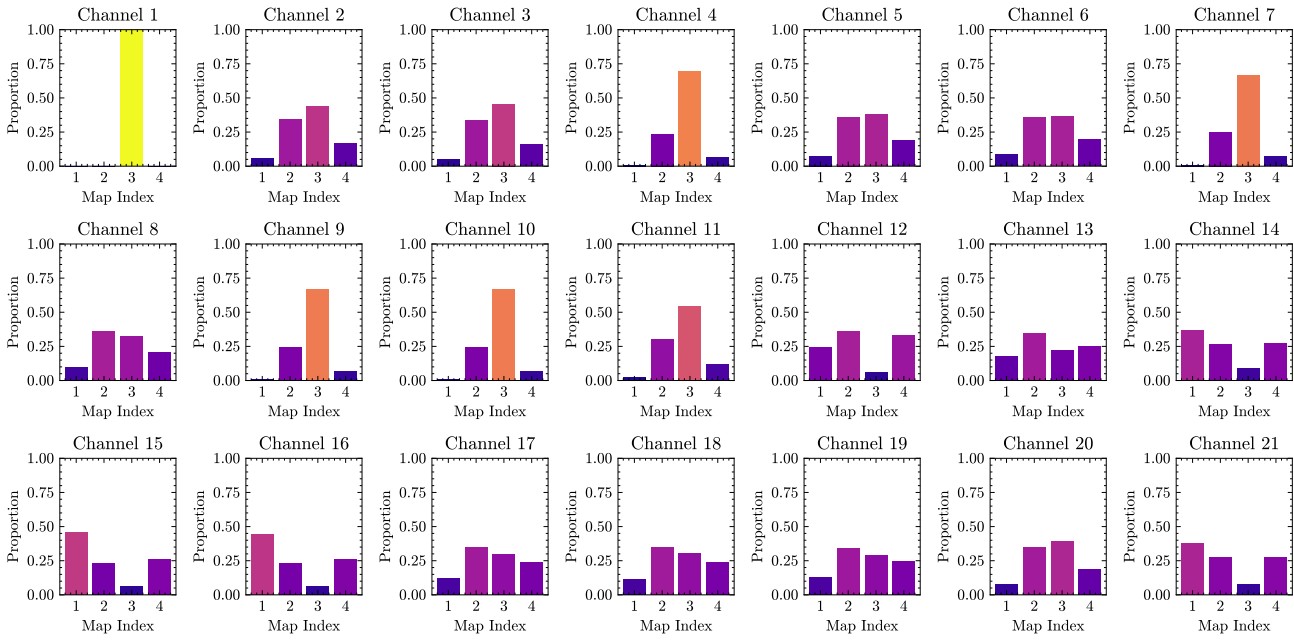

*Figure 13.* Visualizations of mapping allocation situation on all channels.

