# OpenReview forum: "CMoS: Rethinking Time Series Prediction Through the Lens of Chunk-wise Spatial Correlations"
_ICML.cc/2025/Conference — ICML 2025 poster_

### Official Review · Reviewer_SVxE · 2025-03-06

**Overall Recommendation:** 4

**Summary:**

This paper proposes  the CMoS, a highly  lightweight model for time series forecasting tasks. Unlike previous studies, CMoS capture the temporal patterns in a chunk-level manner. The Correlation Mixing mechanism builds robust correlationmatrices, and Periodicity Injection technique help to leverage the periodicity information.

**Claims And Evidence:**

- In figure 1, thios paper claims that "the specific patterns in the time window change greatly, while the spatial correlations of the time series
chunks remain similar." This may be a specific case for time series data with obvious periodicity. This may not hold when time series data shows no periodicity.

**Essential References Not Discussed:**

There are no problems.

**Experimental Designs Or Analyses:**

Yes. There are no problems.

**Methods And Evaluation Criteria:**

Yes.

**Other Comments Or Suggestions:**

No more comments.

**Other Strengths And Weaknesses:**

Strengths:
- The writing is clear and fluent.
- The method is effective and easy to implement.

Weaknesses:
- Some equations exists errors. For example, in the Theorem3.2, the superscript of the summation symbol should be n-1. Similar problem exists in equation 3.
- The details of Periodicity Injection for each dataset should be described in experimental part.
- This is a lightweight model, but SparseTSF seems much more lightweight than this model. Please specify the reason and the advantages against SparseTSF.

**Questions For Authors:**

Please see the weakness and problems above.

**Relation To Broader Scientific Literature:**

Previous methods model the temporal relationship in point-level or learn feeature from patches. This paper proposes learn the relationship in the chunk-level. It provides a new perspective to leverage the relationship between channels, instead of simply employing Channel Independent strategy.

**Theoretical Claims:**

Yes. There are no problems for proofs.

---

> ### Author Rebuttal · Authors · 2025-03-30
>
> **Thank you for your detailed and thoughtful review! We will address your concerns point by point.**
>
> > Q1: The statement may not hold when time series data shows no periodicity.
>
> We appreciate your careful examination of this matter. **Indeed, as you pointed out, the formulation becomes less rigorous for some irregular time series, such as in the case of a time series generated by random walks. Nevertheless, it is notable that periodicity or stationary trend constitutes a fundamental indicator of time series predictability and forecasting potential**. Long-term forecasting can be very challenging if time series exhibit no periodicity or stationary trend. For example, DLinear demonstrates that a naive baseline that simply copies the most recent observation can outperform almost all SOTA deep learning models that on the aperiod financial dataset [1]. We also reach a similar conclusion by analyzing the 1st channel of Weather dataset described in Sec. 5. So, **our statement holds true for time series that are predictable and have strong forecasting potential**. And through extensive experiments, **we demonstrate both the soundness and superiority of our approach, which is designed based on the principles of the statement, across the majority of real-world scenarios**.
>
> *Also, motivated by your concerns, we will include an additional discussion section in the revised paper to elaborate on this issue, as well as other potential limitations.*
>
> > Q2: Some equations exists errors.
>
> Quite thanks for your reminder! We will carefully review all the formulas and correct the errors in the revised paper.
>
> > Q3: The details of Periodicity Injection for each dataset should be described in the experimental part.
>
> Thanks for your valuable suggestions. We will add the below detailed description of Periodicity Injection in our revised paper according to your suggestion:
>
> For each dataset, **firstly, we use the AutoCorrelation Function (ACF) to calculate the dominant period $p$ of this dataset** (the period of all datasets calculated by ACF are listed in the 5th column in Table 7 in Appendix D). Next, during the grid search process for the hyperparameter chunk size, **for each experiment setting, we input the the calculated $p$ and the hyperparameter chunk size $S$ into Algorithm 1 provided in Appendix E, to obtain a modified matrix initialized via Periodicity Injection**. This matrix is then used to replace the first basic correlation matrix of the initialized model, thereby completing the Periodicity Injection operation.
>
> > Q4: Please specify the reason and the advantages against SparseTSF
>
> SparseTSF significantly reduces the number of parameters by downsampling the original time series. Additionally, it adopts a Channel-Independent strategy (Described as One Bus in Appendix B)—modeling a single shared temporal structure for all time series—to minimize the overall number of model parameters. However, the oversimplified modeling of the time series forecasting task results in severely limited representational capacity. **Since it can only model a simple and singular temporal structure for a given time series system, this method fails to accurately capture the diverse temporal patterns that may exist across various time series within the system**. As a result, its forecasting performance tends to be suboptimal.
>
> In contrast, with the help of Correlation Mixing strategy, **CMoS focuses on building several fundamental basic correlations that represent the diverse and inherent patterns of the whole system, and apply a specific mixing strategy for each channel to capture the channel-specific and more accurate temporal structure, thereby greatly improving the prediction performance.** According to the experimental results, CMoS consistently outperforms SparseTSF with a significant margin on datasets containing more than 20 channels (more channels often contain a greater diversity of temporal structures). Therefore, **when comprehensively considering the trade-off between prediction performance and parameter efficiency, our method stands out as the optimal choice among existing approaches including SparseTSF**.
>
> [1] Zeng, Ailing, et al. "Are transformers effective for time series forecasting?." Proceedings of the AAAI conference on artificial intelligence. Vol. 37. No. 9. 2023.
>
> **Thank you again for your kind review, and we hope our response can address your concerns!**

---

> > ### Comment · Reviewer_SVxE · 2025-04-06
> >
> > Thank the authors for their response. I hope these discussions can be incorporated into the paper to enhance its comprehensiveness. My concerns have been addressed, and I am happy to raise my rating to 4.

---

> > > ### Author Response · Authors · 2025-04-06
> > >
> > > Thank you very much! Your suggestions are very helpful for improving the quality of our paper and work, and we will incorporate all the details mentioned in our discussions into the final paper. Once again,  we sincerely thank you for taking the time to review our paper and for raising the rating!

---

### Official Review · Reviewer_fxdG · 2025-03-11

**Overall Recommendation:** 3

**Summary:**

This paper works on time series forecasting task, and the main idea is to split time series into chunks, and build up chunk-chunk spatial correlations to achieve robust time series forecasting. The paper is clearly written, the proposed modules are accompanied with good motivations and solid proofs.

## Update After Rebuttal

I've read the rebuttal and other reviewers' comments, my final rating is weak accept. The reasons why I cannot give higher ratings are: (1) The performance gap in Table 1 is minor; (2) Although the proposed method can greatly reduce the parameter number, the FLOPs and inference time remain comparable to other methods.

**Claims And Evidence:**

1. In Figure 1, which specific techniques are used to obtain the correlation value? It's interesting yet a bit confusing why the obtained correlation values are similar, though the time series in each chunk looks very different. Please give more details.

**Essential References Not Discussed:**

Can you discuss the differences between your method and the following chunk-based techniques?

[1] Ju, Yue, Alka Isac, and Yimin Nie. "Chunkformer: learning long time series with multi-stage chunked transformer." arXiv preprint arXiv:2112.15087 (2021).

[2] Johnsen, Pål V., et al. "Recency-Weighted Temporally-Segmented Ensemble for Time-Series Modeling." arXiv preprint arXiv:2403.02150 (2024).

**Experimental Designs Or Analyses:**

Yes, the conducted experiments are reasonable. A few experiments are missing, and I mention them in "Questions for Authors" below.

**Methods And Evaluation Criteria:**

Yes.

**Other Comments Or Suggestions:**

N.A.

**Other Strengths And Weaknesses:**

Strengths:
1. This paper can effectively reduce the number of model parameters to achieve robust time series forecasting.

Weaknesses:
1. The performance gap is quite minor.

**Questions For Authors:**

1. Can you compare your aggregated spatial correlations (from basic correlations) with attention-based methods? I'm interested if your correlations can be similar or not.
2. In Section 4.4, for the efficiency-related experiments, can you further include time cost and flops? It's possible that you use less parameters but have more computations.
3. Can you provide some failure cases when your method cannot have good performance? Theoretically, I think aggregation-based correlations cannot well cover all cases.

**Relation To Broader Scientific Literature:**

The key contributions of this paper are reducing parameter sizes specially for time series data, I think this technique would be general to all time series related tasks.

**Theoretical Claims:**

Yes, no issues are found.

---

> ### Author Rebuttal · Authors · 2025-03-30
>
> **Thank you for your detailed and thoughtful review! We will address your concerns point by point.**
>
> > Q1: Details about Fig. 1
>
> As an illustrative example, we use the MSE of each pair of chunks as the correlation value (labeled in the figure), and a lower MSE meaning stronger correlation. **Only when the shapes of two chunks are relatively similar can the MSE between them be close to zero, indicating a strong correlation between the two chunks**. In other cases, there is no significant relationship between them since their MSE is usually much greater than zero. We would be pleased if the above description can address your conerns!
>
> > Q2: The differences between your method and other chunk-based techniques.
>
> Thanks for your valuable question. Despite all three approaches employing chunk operations on raw time series,
>
> **both Chunkformer and REWTS  emphasize modeling intra-chunk relationships within individual chunks**. However, this design suffers from poor interpretability, as the internal mechanisms (such as attention weights of intra-chunk correlation) are often opaque. **It's hard for us to understand how these intra-chunk relationships contribute to the future predictions**. Moreover, since intra-chunk modeling is highly sensitive to each value within a chunk, **the model's performance is often greatly affected when the data contains high levels of noise or outliers**.
>
> **In contrast, our CMoS focuses on direclty modeling the relationships between historical chunks and prediction chunks (inter-chunk relationships)**. This approach offers several key advantages:
> - **Enhanced Interpretability**: Our method provides clear insights into how historical chunks influence future predictions (as shown in Sec.5).
> - **Improved Robustness**: The focus on broader temporal relationships rather than fine-grained intra-chunk patterns makes our method more resilient to noise in the data, which is theoretically proved in Sec. 3.1 and experimentally proved in Sec. 4.3.
>
> **We will add this point to our related works to help readers better understand the differences between CMoS and other methods.**
>
> > Q3: Can you compare your aggregated spatial correlations with attention-based methods?
>
> Sure! We visualized the attention-based representation (att1&2.png, based on the attention score of PatchTST's 2 layers) and the correlation of CMoS (cmos.png) on weather dataset in anonymous repo https://anonymous.4open.science/r/kjUH.
>
> From the picture, we find that there is a significant difference between the two. The attention-based correlation represents the attention relationships **among historical data points, but it does not reveal how these relationships contribute to the prediction of future time series**. As a result, it is difficult to derive an intuitive explanation of the forecasting process from the attention-based correlation. In contrast, our correlation **directly captures the mapping from historical data to the future time series**. This allows us to **clearly see which historical chunks contribute more to the prediction of future chunks**, making the model’s decision process more interpretable.
>
> > Q4: Can you further include time cost and flops?
>
> That's a quite good suggestion! We provide the inference FLOPs, GPU memory footprint, and inference time of CMoS and other baselines on Electricity dataset using a 3090 GPU as follows. The batch size of all methods is set to 64.
>
> ||FLOPS|Memory|Infer. Time|
> |-|-|-|-|
> |Dlinear|5.31G|245MB|1.81s|
> |CycleNet|5.68G|267MB|1.83s|
> |SparseTSF|1.02G|262MB|1.49s|
> |FITS|5.33G|691MB|4.71s|
> |iTransformer|249.51G|2271MB|1.92s|
> |PatchTST|1196.08G|22014MB|2.90s|
> |TimeMixer|10.58G|18642MB|2.85s|
> |CMoS|2.96G|252MB|1.58s|
>
> Although the memory allocation strategy and powerful computational performance of the 3090 may narrow the gap in computational overhead among models, it can still be seen that, **CMoS consistently maintains an advantage in computational overhead** except for SparseTSF. It is also notable that the prediction performance of CMoS greatly outperforms SparseTSF, especially on those datasets with more channels. **This means CMoS can achieve the best effectiveness-efficiency balance among all methods**.
>
> > Q5: Some failure cases.
>
> When the underlying data distribution shifts significantly over time (i.e. concept drift), such as sudden changes in market behavior or consumer patterns, the basic correlations may not include the new time strutures, so these rapid distribution shifts can affect the prediction accuracy of CMoS.
>
> However, it's also important to note that this is a fundamental difficulty in time series forecasting that the entire field is actively working to address. A possible solution is to quickly update the model when facing concept drifts, and owing to the efficiency advantage, CMoS can be updated more rapidly and at a higher frequency. **This rapid adaptation capability allows CMoS to mitigate the impact of concept drift more effectively than other methods**.

---

> > ### Comment · Reviewer_fxdG · 2025-04-02
> >
> > Thanks for your nice rebuttal, I do not have other questions, I keep my score as weak accept. The reasons why I cannot give higher ratings are: (1) The performance gap in Table 1 is minor; (2) Although the proposed method can greatly reduce the parameter number, the FLOPs and inference time remain comparable to other methods. Taking all these factors into consideration, the motivation, i.e., reduce parameter cost, appear to be not that strong.

---

> > > ### Author Response · Authors · 2025-04-02
> > >
> > > # ==Update==
> > > Dear reviewer,
> > >
> > > **We are delighted to share our newest experiments on model efficiency with you!** To further investigate the model’s real-world practicality, we conducted additional experiments **by disabling the GPU and using only a single CPU core**, in order to simulate the model's performance **on edge devices with limited computational resources**, and there inference time under this limitation is listed in below:
> > >
> > > |Method|Infer. Time (One CPU)|
> > > |-|-|
> > > |DLinear|45.82s|
> > > |CycleNet|51.17s|
> > > |SparseTSF|16.84s|
> > > |FITS|72.34s|
> > > |iTransformer|393.09s|
> > > |PatchTST|2676.03s|
> > > |TimeMixer|1512.43s|
> > > |CMoS|25.23s|
> > >
> > > It can be observed that **the inference time of CMoS is at least 40% shorter than that of existing methods** except for SparseTSF (while CMoS has great performance advantages compared with SparseTSF). This indicates that CMoS has a significant computational advantage, making it suitable for deployment on a wider range of edge devices for high-quality time series forecasting.
> > >
> > > **With regard to your another concern about minor performance gap**, while CMoS may not demonstrate very large margins of improvement over the second-best method on individual datasets, **it uniquely maintains state-of-the-art performance consistently across multiple datasets**, a characteristic not shared by any baseline method. **Unlike other approaches that might excel on specific datasets but show inconsistent performance across different scenarios, CMoS exhibits robust and superior performance across a diverse range of datasets**. This consistent excellence across multiple benchmarks underscores the versatility and reliability of our approach, especially considering that CMoS is a super-lightweight method.
> > >
> > > *We’d greatly appreciate it if our response addresses your remaining concerns! And we are glad to engage in continued discussion with you!*
> > >
> > > # ==Previous reply==
> > > Dear reviewer, thank you for your timely response.
> > >
> > > Due to the super-lightweight design of our model, it is far from fully utilizing the computational resources of high-performance GPUs. As a result, our efficiency metric may not show a very great advantage over existing methods. However, for edge devices with strict memory constraints, the number of parameters directly determines whether the algorithm can be deployed on such devices. This is of great practical significance for real-world applications.
> > >
> > > Inspired by your suggestion, we plan to develop an ONNX version of CMoS to facilitate efficient time series forecasting on edge devices, and provide more possibilities for the broader application of CMoS and other further light-weight works.
> > >
> > > Once again, we sincerely thank you for taking the time to review our paper and providing these valuable suggestions! We will carefully revise our paper following your suggestions.

---

### Official Review · Reviewer_VtAP · 2025-03-13

**Overall Recommendation:** 3

**Summary:**

This paper presents CMoS, a super-lightweight time series forecasting model that utilizes chunk-wise spatial correlations to achieve parameter-efficient and interpretable predictions. The key innovation lies in directly modeling the spatial dependencies between time blocks of fixed size, rather than point-oriented patterns, as theory and experience suggest, to enhance noise robustness. CMoS introduces a correlation hybrid strategy that combines a small group of shared basis correlation matrices (e.g., long-term, short-term, periodic) with channel-specific adaptive weights to achieve diversified time structure modeling while maintaining minimal parameters. In addition, periodic injection through weight initialization accelerates the convergence of periodic data. The experiment demonstrated state-of-the-art performance across seven benchmarks, with interpretable correlation matrices revealing different time dependencies (e.g., daily periodicity, residual trends). This work highlights the inherent simplicity of time structures and provides a framework for resource efficiency forecasting.

## update after rebuttal
I thank the authors for their thorough response to reviewers' feedback and the improvements made to the paper. After examining the rebuttal and considering the other reviews, I've updated my overall recommendation to "3 - Weak accept" as the authors have sufficiently addressed the primary questions and concerns I raised.

**Claims And Evidence:**

Yes, I believe the paper's claims are supported by clear and convincing evidence.

**Essential References Not Discussed:**

I believe there is no essential reference missing from the discussion.

**Experimental Designs Or Analyses:**

Experiments are valid using standard benchmarks and metric. Ablation studies support key designs (chunk-wise modeling, correlation mixing), and parameter efficiency is well quantified.

**Methods And Evaluation Criteria:**

The proposed method is reasonable for time series forecasting. With the support of theoretical analysis and ablation research, Chunk-wise spatial correlation modeling solves the problem of noise robustness and parametric efficiency. The correlated mixing strategy effectively balances model capacity and lightweight design through  shared basis matrices, while allowing for channel-specific adaptations. Periodicity injection provides a practical inductive bias for cyclical patterns without overcomplicating the architecture. Standard Indicators (MSE/MAE) are used to assess seven established benchmarks to ensure fair comparisons.

**Other Comments Or Suggestions:**

No other comments.

**Other Strengths And Weaknesses:**

Strengths:
1.	The paper is well-structured, with clear technical exposition (e.g., chunk-wise formulation, correlation mixing pseudocode) and intuitive visualizations (Fig. 7-8) that enhance interpretability.
2.	The paper compellingly critiques the limitations of channel-independent strategies for lightweight models and introduces correlation mixing as a novel middle ground between parameter efficiency and multi-pattern modeling.
3.	The extreme parameter efficiency (1% of DLinear) and interpretable correlation matrices offer direct value for edge deployment and domain analysis (e.g., energy systems).

Weaknesses:
1.	Theorem 3.2 assumes linear and Gaussian noise, ignoring nonlinear dependencies and real-world noise types (e.g., burst noise), limiting its practical relevance.
2.	Despite the emphasis on lightweight design, key metrics such as inference speed, and memory footprint are ignored, leaving the utility of the deployment unproven.
3.	Chunk size selection depends on prior periodic knowledge (e.g., divisor of 24/168), requiring manual tuning of the new data set, and reducing the availability of aperiodic or irregularly sampled series.

**Questions For Authors:**

1.	How could Theorem 3.2 be extended to nonlinear dependent or non-Gaussian noise (e.g., burst noise), and what empirical evidence supports its robustness under such conditions?
2.	Despite the emphasis on lightweight design, why are inference speed and memory footprint excluded from the evaluation? Can you provide some benchmarks to verify the usefulness of your deployment?
3.	What strategies can automatically select the chunk size of the dataset without explicit periodic or irregular sampling, thereby reducing the reliance on manual tuning?

**Relation To Broader Scientific Literature:**

CMoS builds on recent advances in lightweight time series models (e.g., DLinear, FITS) but uniquely addresses their limitations in handling diverse temporal structures. While channel-independent strategies (PatchTST, SparseTSF) reduce parameters but restrict model capacity, and channel-mixing methods (iTransformer) incur high complexity, CMoS bridges this gap via correlation mixing—sharing basis matrices while adapting to channel-specific patterns, akin to "mixture of experts" but tailored for spatial correlations. Its chunk-wise modeling aligns with patching in PatchTST but prioritizes robustness over semantic embeddings. Periodicity injection extends CycleNet’s explicit cyclical modeling but integrates it into interpretable correlation weights. The work advances the paradigm of "simple yet effective" models, demonstrating that lightweight designs can achieve both efficiency and expressiveness by rethinking temporal dependencies.

**Theoretical Claims:**

The paper’s theoretical claim (Theorem 3.2) asserts that chunk-wise spatial correlations reduce noise sensitivity compared to point-wise modeling. The proof in Appendix F correctly applies the Cauchy-Schwarz inequality, showing that averaging the point-wise weights into chunk-wise weights reduces the L2 norm of the parameter and thus the noise variance. However, the theorem assumes linear and Gaussian noise, which may not be exactly consistent with real-world time series dynamics (e.g., non-Gaussian noise, nonlinear dependence). While the proof is mathematically sound under these assumptions, its practical significance depends on how closely the linear chunk-wise model approximates the real data.

---

> ### Author Rebuttal · Authors · 2025-03-30
>
> **Thank you for your detailed and thoughtful review! We will address your concerns point by point.**
>
> > Q1: How could Theorem 3.2 be extended to nonlinear dependent or non-Gaussian noise (e.g., burst noise), and what empirical evidence supports its robustness under such conditions?
>
> - **Extended to Non-Gaussian noise like burst noise**
>
> Take the burst noise as an example, it can be viewed as extreme deviations that occur in the tail of a noise distribution. So we can define burst noise $B(t)$ mathematically as follows based on extreme value theory. Let $X(t)$ be a noise process. According to the Pickands–Balkema–de Haan theorem, the conditional distribution of the exceedances events exceeding a sufficiently high threshold $u$ follow a Generalized Pareto Distribution (GPD): $GPD(y ; \sigma, \xi)= P(Y(t)\le y|X(t)>u) \approx
> 1-\left(1+\frac{\xi y}{\sigma}\right)^{-1 / \xi}$,where $y=x-u>0$, $\sigma>0$ is the scale parameter, and $\xi > 0$ is the shape parameter. Next, since the burst noise can be regarded as discrete events, we define their occurence times ${T_i}$ follow a Poisson process with intensity $\lambda(u)$. So we can finally define burst noise $B(t)$ as $B(t) = y+u,\text{if}\ t\in\{T_i\},\ \text{otherwise}\ 0$, where $y$ is i.i.d. $GPD(\sigma, \xi)$.
>
> However, when we attempt to replace $\delta$ with $B$ in Definition 3.1, we find it difficult to theoratically analyze $Var(\theta^TB)$ since **B is sparse and thus $Var(\theta^TB)$ relies heavily on specific samples of GPD**.
>
> As an alternative, we choose to **experimentally compare the performance** of chunk/point-level modeling on the time series with random burst noise. Specifically, we construct several time series using sine functions, and following the above formulation, we additionally inject some burst noises into all time series. Some segments of these time series are visualized in *burst.png* in the anonymous repo https://anonymous.4open.science/r/kjUH. The prediction results are listed in the following table, and **we can conclude that chunk-level modeling is more robust than point-level modeling when facing burst noise, which can be seen as an empirical extension of Theorem 3.2 on other non-Gaussian noise like burst noise**.
> ||MSE|MAE|
> |-|-|-|
> |Chunk-level|0.0246|0.1166|
> |Point-level|0.0249|0.1176|
>
> - **Extended to nonlinear dependency**
>
> With regard to **modeling nonlinear dependency**, the presence of nonlinear interactions makes it difficult to derive closed-form solutions or establish rigorous mathematical proofs. Specifically, in Definition 3.1, if $f(\theta)$ is a nonlinear function, it's hard to obtain a parametric mathematical expression for $Var(f(x';\theta)-f(x;\theta))$, which hinders the following theoretical derivations. So we design a 3-layer MLP with **nonlinear activation** to validate the effectiveness of chunk-level modeling. From the results, **we can find that chunk-level modeling outperforms point-level modeling in most cases, indicating its robustness on multiple datasets**.
> ||Ele.|Tra.|Wea.|ETTh1|ETTh2|ETTm1|ETTm2|
> |-|-|-|-|-|-|-|-|
> |Chunk NonLinear|0.166|0.414|0.237|0.410|0.362|0.357|0.263|
> |Point NonLinear|0.170|0.431|0.245|0.413|0.381|0.359|0.260|
>
> > Q2: Inference speed and memory footprint
> >
>
> That's a quite good suggestion! We provide the inference FLOPs, GPU memory footprint, and inference time of CMoS and other baselines on Electricity dataset using a 3090 GPU as follows. The batch size of all methods is set to 64.
>
> ||FLOPS|Memory|Infer. Time|
> |-|-|-|-|
> |Dlinear|5.31G|245MB|1.81s|
> |CycleNet|5.68G|267MB|1.83s|
> |SparseTSF|1.02G|262MB|1.49s|
> |FITS|5.33G|691MB|4.71s|
> |iTransformer|249.51G|2271MB|1.92s|
> |PatchTST|1196.08G|22014MB|2.90s|
> |TimeMixer|10.58G|18642MB|2.85s|
> |CMoS|2.96G|252MB|1.58s|
>
> Although the memory allocation strategy and powerful computational performance of the 3090 may narrow the gap in computational overhead among models, it can still be seen that, **CMoS consistently maintains an advantage in computational overhead** except for SparseTSF. It is also notable that the prediction performance of CMoS greatly outperforms SparseTSF, especially on those datasets with more channels. **This means CMoS can achieve the best effectiveness-efficiency balance among all methods**.
>
> > Q3: Automatically select the chunk size of the dataset.
>
> That's quite a valuable question. As there is no enough information about period, using adhoc chunk size might not be a good choice. The good news is that **since CMoS is a super-lightweight model, it is entirely affordable performing hyperparameter search within a certain range in most cases. Therefore, in practice, we can take advantage of advanced hyperparameter optimization algorithms (such as Bayesian optimization) or frameworks (such as Optuna) to automatically find the optimal chunk size for best performance** on these aperiodic or irregularly sampled series.
>
> *Thank you again for your valuable review, and we hope our response can address your concerns.*

---

### Official Review · Reviewer_aVfa · 2025-03-15

**Overall Recommendation:** 3

**Summary:**

There's a recent line on making small architectures that match the performance of large deep learning models for TS forecasting, which raises the question of the relevancy of DL for time series forecasting. The authors propose CMoS, a novel architecture for time series forecasting that is very lightweight. The main contributions of the architecture include two elements: (1) chunk-wise spatial correlation modelling, which models the prediction of each chunk as a linear combination of previous chunks, and (2) correlation mixing, which uses cross-channel aggregation to get channel-specific spatial correlation mixtures with a low parameter count. The authors also introduce periodicity injection due to the structure of the correlation matrix, allows them to introduce pre-defined periodic peaks into the spatial correlation matrix's initialization. The authors use standard MSE loss and RevIN. Experiments are run on typical TSLib datasets against a few existing baselines. The authors ablate the different components of their proposed architecture, and offer analyses of their architecture's efficiency and interpretability.

**Claims And Evidence:**

- Novel architecture CMoS
  - chunk-wise Spatial correlation modelling
  - Correlation mixing strategy

Using convnets for forecasting is not a new idea. However, the related work lacks any reference to these works. As such, it is difficult to determine how novel the introduced architecture actually is, and what components of it are novel. As the authors state, patching and chunking are quite similar, but the discussion on this topic is quite brief. As is stands, it is difficult to evaluate the novelty of the claims without a proper literature review on comparable architectures. Part of the purpose of the Related Work section is to help differentiate works that are similar in nature, and so should discuss the use of patching more extensively, any differences it has with chunking (other than purpose), the use of convolutions for forecasting. For example, see https://arxiv.org/abs/1906.04397, section 2, paragraph 2 which discusses many other examples of architectures that leverage similar convolutional biases.

- Novel weight init strategy for periodicity injection

This is an interesting strategy for smart initialization. One obvious limitation to this method is not only that you must know what the periodicity looks like, that periodicity must be fixed (cannot vary over time). Furthermore, it is unclear what happens if you inject a periodicity that is actually incorrect into a model, if e.g. you make an incorrect assumption around the periodicity. I would like to see experiments around this. Nevertheless, I find this idea quite interesting, albeit hard to generalize to other architectures.

- CMoS is SOTA

The baselines that the authors compare to are those available in the tslib library. Therefore, if their results are better than those of the best model in that library, they are SOTA. Looking at the best model on there currently, CMoS is SOTA on those datasets. However, Tslib seems to be missing many models, including Time-LLM, which performs better (see Table 1 of https://arxiv.org/abs/2310.01728). It would also be useful to add some non-DL baselines to compare, e.g. naive, naive with drift, seasonal naive. Also, if you're using TFB, why not have included crossformer in your table? PatchTST is often second-best, and crossformer is competitive with patchTST in the TFB paper.

- Interpretable learned spatial correlation matrices

The authors visualize the spatial correlation matrices and interpret them for the weather dataset, which is informative.



- Difference between chunk and patch: really underexplored. Describe in an appendix or something?

**Essential References Not Discussed:**

- The review needs to go back further than 2023, especially considering that this is a more traditional paper (see "Claims And Evidence", the first claim, the paper on temporal convolutions, the related work section, paragraph 2, as a place to start).

**Experimental Designs Or Analyses:**

The datasets are widely used and typical within the field.

- It's unclear how you chose to specify the grids for the chunk size and spatial correlation search. It's also unclear is what is meant by "lookback window search".

- The authors conducted ablations on the components of the method, showing they are all important for the method's success.

- Interpretability analysis through visualizations. Interesting discussion of the mappings. A way to quantify this approach would be better, especially if you could tie these mappings to the actual time series. Figure 8 is a start, and it would be interesting to expand this analysis to other datasets, or to compare datasets where some transfer might be expected and see if the mappings have similarities across datasets, e.g. in between the ETT datasets.

**Methods And Evaluation Criteria:**

See above re: CMoS being SOTA.

**Other Comments Or Suggestions:**

N/A

**Other Strengths And Weaknesses:**

Originality: This work further elucidates the importance of architectural choices for efficient ts forecasting.

Significance: The results of this work are significant, in that they show solid performance with less parameters than typical methods.

Clarity: The paper structure is standard and clear. It's well-written overall, with a few typos.

**Questions For Authors:**

- Did you ablate for overlapping chunks?
- How about multiple layers of convolution?
- Why are the K correlation matrices shared by all channels?
- how do you select the degree of smoothing to apply during the two-stage weight allocation?
- Are the train/test splits also across time?
- Can the periodicity injection be applied to other architectures, for example, those with nonlinearities?

**Relation To Broader Scientific Literature:**

This paper relates to a line of work around deep learning for time series forecasting. Autoformer introduced the Time Series Library https://github.com/thuml/Time-Series-Library that has been used regularly by the community to benchmark TSF models on a set of common datasets. Among these models, some are extremely parameter-efficient and question whether transformers are useful/necessary for time series forecasting, such as DLinear https://arxiv.org/abs/2205.13504. This work continues that line of work, using the same TSLib to benchmark their model and comparing against baselines in that repo.

**Theoretical Claims:**

Theorem 3.2 Has a proof in Appendix F, that I did not check.

---

> ### Author Rebuttal · Authors · 2025-03-30
>
> **Thank you for your detailed and thoughtful review! We will address your concerns point by point.**
>
> > Q1: Novelty of some components
>
> Your example is quite helpful! So we explain the two aspects you mentioned following the format you provided:
> - The use of ConvNet.
>
> Existing methods like DeepTCN or TimesNet employs one ConvNet backbone for all channels. Since different channels may exhibit different noise levels, only one backbone may struggle to resist interference from various levels of noise. In contrast, **CMoS allocate a specific lightweight ConvNet for each channel to eliminate the channel-specific noise variations**, thus enhancing the model's robustness.
>
> **It is also notable that compared to this specific component, the whole Correlation-Mixing framework (including ConvNet) is a more important innovation.** As shown in Appendix B, **compared with existing channel strategies, our Correlation-Mixing can effectively model diverse temporal and even cross-channel dependencies with great efficiency**. ConvNet plays an important role in reducing the effect of noise in this framework.
> - Chunk v.s. Patching.
>
> Technically, patching **only splits the historical series into segments**, and patch-based models like PatchTST focus on **generating aggregated representation of the correlations between these historical segments** (similar to the high-level semantic information in LLMs) and then decode the representation to future time points. The black-box nature of such representations make it hard to figure out how specific segment influence the final prediction, limiting the interpretability of these methods.
>
> In contrast, chunk **splits both historical and future series**, and instead of learning the high-level representations, chunk-based CMoS focus on **directly modeling of the spatial correlation between historical and future segments**. which is quite interpretable. Also, we proved that chunks bring benefits in robustness and efficiency.
>
> *To better understand their differences in interpretability*, we visualized the patch-based representation (att1&2.png, based on the attention score of PatchTST's 2 layers) and chunk-based correlation (cmos.png) on weather dataset in anonymous repo https://anonymous.4open.science/r/kjUH. We can easily figure out how historical segments contribute to future segments for each time series in cmos.png ,while it's hard to obtain similar or other interpretable information from att1&2.png.
>
> **We will provide more details and recent works in the next version according to your valuable suggestions!** Also, reviews about works before 2023 will be added.
> > Q2: Inject incorrect periodicity
>
> That's an interesting question. To simulate this case, we randomly select an integer x!=period as interval in injection phase. Comparing the below results (MSE) with Table 3 in the paper,  it can be seen that **using incorrect period performs barely better than random initialization**.  So we suggest injecting the correct period only when most time series in the system passed the ACF test—a scenario that commonly occurs in real-world applications.
> ||Ele.|Tra.|Wea.|ETTh1|ETTh2|ETTm1|ETTm2|
> |-|-|-|-|-|-|-|-|
> |wrong period|0.130|0.372|0.151|0.371|0.297|0.294|0.174|
> > Q3: More Baselines
>
> We follow your suggestion to include more baselines. The prediciton results (MSE) of naive, seasonal naive, Time-LLM and Crossformer are listed in the below. *Since Time-LLM leverage very large LLM as part of the model, the trainging time is quite long (over 2 days for single setting). Therefore, we currently have only obtained partial results. The full results are on in its way.*
>
> From the results, CMoS outperforms these methods, indicating that CMoS is a quite effective model.
> ||Ele.|Tra.|Wea.|ETTh1|ETTh2|ETTm1|ETTm2|
> |-|-|-|-|-|-|-|-|
> |Naive|1.611|2.770|0.353|1.319|0.533|1.271|0.385|
> |Naive Season|0.230|0.630|0.371|0.598|0.477|0.489|0.358|
> |Crossformer|0.181|0.523|0.235|0.452|0.861|0.465|0.589|
> |Time-LLM|-|-|-|0.414|0.340|0.360|0.262|
> |CMoS|0.158|0.396|0.220|0.403|0.331|0.354|0.259|
> > Q4: Other concerns
>
> Very sorry that the response here cannot cover every point you have mentioned due to the text length limitation, and **we can further discuss any uncovered points during the discussion phase**. Here are the brief replies for some points:
> - Parameter concerns: We determined the grid sets through extensive experiments.
> - More visualizations and ablations: These are valuable suggestions. We will perform more experiments and  analysis.
> - K matrices are shared: As mentioned in *Sec. Introduction*, the K matrices are designed to learn some basic temporal structures in the system, and each channel finds a specific way to combines these basic correlations. This design brings both efficiency and robustness benefits.
> - Nonlinear injection: If certain modules strongly correlate with periodicity, we can generalize Periodicity Injection to these modules. We believe this would be an interesting direction to explore.

---

### Decision · Program_Chairs · 2025-05-01

**Decision:**

Accept (poster)

**Comment:**

This paper presents CMoS, a lightweight time series forecasting model that introduces chunk-wise spatial correlation modeling and correlation mixing to achieve strong performance with minimal parameters. The idea of modeling inter-chunk correlations, along with a shared-and-adapted correlation strategy across channels, is both elegant and practical—offering a strong balance between efficiency, interpretability, and predictive accuracy. Reviewers appreciate the thoughtful design, clear writing, and the extensive experimental evidence demonstrating consistent performance across multiple benchmarks. While some concerns were raised about the marginal performance gains over recent baselines and the reliance on prior knowledge for periodicity injection, the authors addressed these issues reasonably well in the rebuttal. Overall, this paper makes a valuable and well-executed contribution to the growing body of work on efficient and interpretable time series forecasting. I lean toward acceptance.